# The influence of zooplankton and oxygen on the particulate organic carbon flux in the Benguela Upwelling System

Luisa Chiara Meiritz [1,4], Tim Rixen [1,2], Anja Karin van der Plas[3], Tarron Lamont [5,6,7], Niko Lahajnar [1]

[1] Universität Hamburg, Institute of Geology, Hamburg, Germany
[2] ZMT Leibniz-Centre for Tropical Marine Research, Bremen, Germany
[3] NatMIRC Ministry of Fisheries and Marine Resources, Swakopmund, Namibia
[4] GEOMAR Helmholtz Center for Ocean Research, Kiel, Germany
[5] Oceans & Coasts Research, Department of Forestry, Fisheries, and the Environment, Cape Town, South Africa
[6] Bayworld Centre for Research & Education, Cape Town, South Africa
[7] Oceanography Department, University of Cape Town, Cape Town, South Africa

*Correspondence to*: Luisa Chiara Meiritz (lmeiritz@geomar.de/ luisa-c-meiritz@web.de)

**Abstract.** We conducted sediment trap experiments in the Benguela Upwelling System (BUS) in the south-eastern Atlantic Ocean to study the influence of zooplankton on the flux of particulate organic carbon (POC) through the water column and its sedimentation. Two long-term moored and sixteen short term free-floating sediment trap systems (drifter systems) were deployed. The mooring experiments were conducted over more than a decade (2009-2022) and the sixteen drifters were deployed on three different research cruises between 2019 and 2021. Zooplankton was separated from the trapped material and divided into 8 different zooplankton groups. In contrast to zooplankton which actively carries POC into the traps in the form of biomass (active POC flux), the remaining fraction of the trapped material was assumed to fall passively into the traps along with sinking particles (passive POC flux). Our results show, in line with other studies, that copepods dominate the active POC flux, with the active POC flux in the southern BUS (sBUS) being about three times higher than in the northern BUS (nBUS). In contrast, the differences between the passive POC fluxes in the nBUS and sBUS were small. Despite large variations, which reflected the variability within the two subsystems, the mean passive POC fluxes from the drifters and the moored traps could be described using a common POC flux attenuation equation. However, the almost equal passive POC flux, on the one hand, and the high POC concentration in the surface sediments of the nBUS in comparison to the sBUS, on the other hand, imply that the intensity of the near-bottom oxygen minimum zone (OMZ), which is more pronounced in the nBUS than in the sBUS, controls the preservation of POC in sediments significantly. This highlights the contrasting effects of the globally observed expansion of OMZs, which on the one hand mitigates the accumulation of $CO_2$ in the atmosphere and the ocean by increasing POC storage in sediments and on the other hand poses a threat to established ecosystems and fisheries.

## Introduction

Carbon storage by pelagic marine ecosystems, known as the biological carbon pump, exerts a strong control over atmospheric $CO_2$ concentrations by influencing $CO_2$ storage in the ocean and underlying sediments. Although scientific studies have widely shown that the biological pump responds to climate change (e.g., Devries and Deutsch, 2014; Duce et al., 2008; Laufkötter et al., 2017; Riebesell et al., 2007) and is affected by fisheries (Bianchi et al., 2021), it is not yet possible to predict the extent and the signs of changes (Laufkötter and Gruber, 2018;

Passow and Carlson, 2012; Rixen et al., 2024). These uncertainties reduce confidence in climate predictions (Passow and Carlson, 2012), call into question sustainability criteria related to the growing blue economy (Jouffray et al., 2020) and assessments of the state of the ocean such as the Ocean Health Index (Halpern et al., 2012). In addition, pelagic ecosystems, which fuel the biological carbon pump and enable the transfer of POC to the

sediment, are not considered as blue carbon ecosystems (e.g. Lovelock and Duarte, 2019; Macreadie et al., 2019), which means that their response to human perturbation is largely ignored in national reports to the UNFCC (United Nations Framework Convention on Climate Change) in the framework of the Paris Agreement. As shelves located in the 200-mile exclusive economic zone (EEZ) are of great relevance for the global carbon cycle (Rixen et al., 2024), lately efforts are being made to include sediments in the blue carbon concept (European Marine

Board, 2023; von Maltitz et al., 2024). To emphasize the relevance of this effort, it should be noted that the carbon storage in the EEZs with 1092 - 1166 Pg C (Atwood et al., 2020) by far exceed those in the classic blue carbon ecosystems (salt marshes, mangroves and seagrasses: ~7.3 - 22.7 Pg C (Pendleton et al., 2012).

One of the difficulties in studying pelagic ecosystems is that they evolve in a moving medium, the ocean. The residence time of ocean water on the shelf is on average around 12 to 17 months (Lacroix et al., 2021), so that it

falls only temporarily under the jurisdiction of an individual state. Nevertheless, they have a long-term impact on carbon storage in territorial waters through their influence on carbon sedimentation. Although processes that control the delivery of POC to the sediment are known in general, there are also a number of unknown and difficult to determine processes as well as methodological problems in determining POC fluxes to the sediment. As a result, global estimates of the amount of POC exported from the sunlit surface ocean (export production) vary between

1.8 - 27.5 $10^{15}$ g C $yr^{-1}$ (Del Giorgio and Duarte, 2002; Honjo et al., 2008; Lutz et al., 2007).

The delivery of POC to the sediment begins with primary production, which converts dissolved inorganic carbon into POC. Incorporated into particles and following gravity, these particles sink through the water column onto the sediment. Three different types of particles are generally distinguished, namely faecal pellets from zooplankton and fish, amorphous aggregates (marine snow) and remains of dead marine organisms (Turner, 2015). Global

modelling studies conclude that faecal pellets are on average responsible for 16 - 85 % of gravitational POC export (Archibald et al., 2019; Nowicki et al., 2022), which is consistent with sediment trap results. In sediment trap samples, the proportion of faecal pellets in the POC can vary between 1 and 100 %, but is often around 40 %. (Turner, 2015).

On their way through the water column, bacteria and zooplankton degrade the sinking organic material. The

decrease in sinking POC with depth has been described by a whole series of relatively simple attenuation equations, of which the so-called Martin equation is probably the best known (Martin et al., 1987). The Martin equation (Eq. 1) is based on the POC flux at the base of the mixed layer ($F_{MLD}$), the water depth (z), the mixed layer depth (MLD), and the attenuation rate (b).

$$F_z = F_{MLD} * (\frac{z}{MLD})^b \qquad \textbf{(1)}$$

The POC flux at the base of the MLD corresponds to the export production which, like all other parameters, can vary spatially and temporally (Martin et al., 1987 and e.g. Giering et al. 2014). A comparison of some of these equations shows that they all represent the measured data well but differ significantly in their implications, e.g., with regard to the calculation of the POC degradation rates inherent in them (Cael and Bisson, 2018). In addition, these equations have also been criticized because they do not consider the role of zooplankton. The diurnal vertical

migration of zooplankton is one of the largest known mass movements in the animal kingdom, extending from the surface to water depths of 200 to 650 m (Bianchi et al., 2013). However, model calculations show that zooplankton

degrade approximately 15 % and 43 % of the exported POC (Bianchi et al., 2013; Archibald et al., 2019). On average, this would be around 30 %, which is consistent with the results of a comprehensive field study from Giering et al. (2014) in the Atlantic Ocean. This study found that 70 - 92 % of the sinking POC at depths between 100 and 1000 m was degraded by bacteria, which conversely means that up to 30 % of the POC is decomposed by zooplankton (Giering et al., 2014). To investigate the influence of zooplankton on POC flux and sedimentation, we conducted sediment trap experiments in the BUS, where the accumulation of POC in the sediment can become so high that in some places 'mud belts' are formed, characterized by POC concentrations >12 % and high POC storage in the sediments (Fig. 1, van der Plas et al., 2007; Monteiro et al., 2005; Emeis et al., 2018; Atwood et al., 2020).

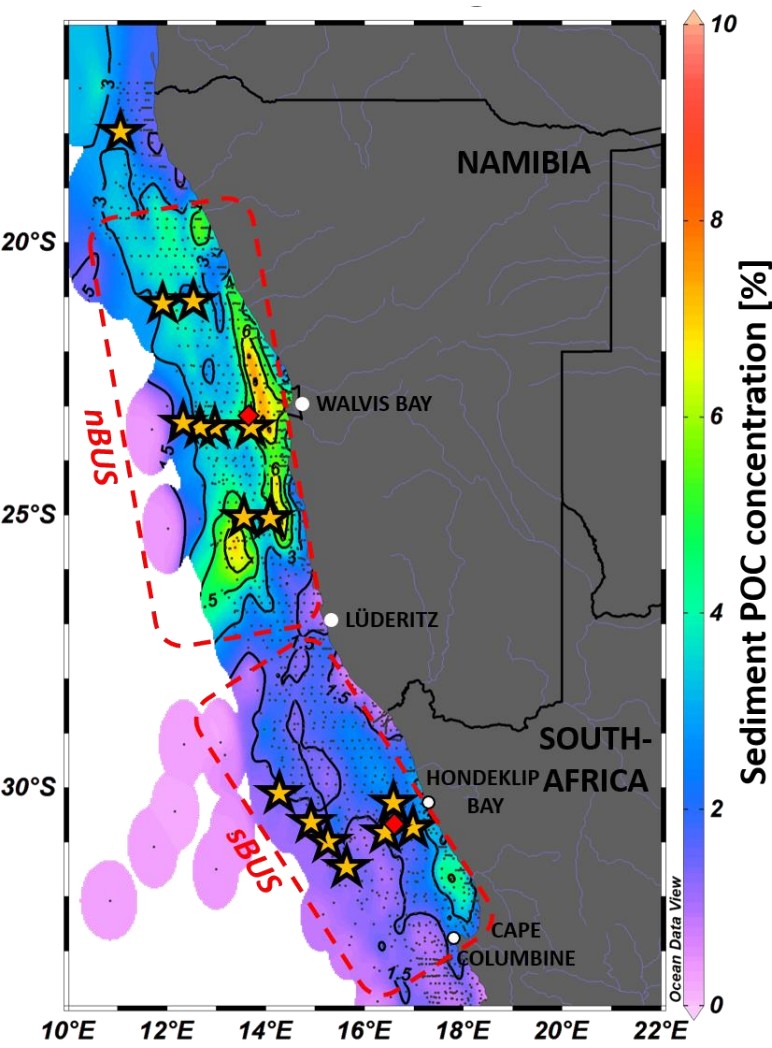

**Figure 1** POC concentration of surface sediment samples in the BUS from various research expeditions published in Emeis et al. (2018). Yellow stars show deployment drifter positions; red diamonds show long-term mooring locations. Contour lines in 1.5 % steps. Red dashed areas indicate the nBUS and sBUS, respectively (Hutchings et al., 2009).

**Working area**

The BUS is one of the four major Eastern Boundary Upwelling Systems, which are among the most productive marine ecosystems in the world's ocean (Chavez and Messié, 2009; Carr, 2001). Although they cover only 2 % of the global ocean surface, they provide more than 20 % of the total global marine fishery yields (Pauly and Christensen, 1995; Sydeman et al., 2014) and contribute about 11 % to the global export production (Chavez and

Toggweiler, 1995). In almost all Eastern Boundary Upwelling Systems, distinct OMZs have been formed below the euphotic zone at depths between approximately 100 and 1000 m (e.g. Monteiro et al., 2011). They are the product of high oxygen consumption, caused by the degradation of the exported POC, compared to the ventilation of the OMZ (Rixen et al., 2020 and references therein). Their expansion due to global warming is considered one of the greatest threats to marine life (Stramma et al., 2008; Stramma et al., 2012), alongside global warming and ocean acidification (https://www.globalgoals.org/goals/14-life-below-water/).

The BUS extends approx. from the Kunene river (~17°S) in the north to Cape Agulhas (~35°S) in the south. It is driven by the southeast trade winds that result from the pressure difference between the South Atlantic high and the continental low over southern Africa, which leads to the formation of individual, particularly prominent upwelling cells along the shoreline (e.g. Kämpf and Chapman, 2016; Sell et al., 2024; Shannon and Nelson, 1996; Veitch et al., 2009). The strongest is the Lüderitz Cell, which divides the BUS at about 27°S into a northern (nBUS) and a southern subsystem (sBUS) (Hutchings et al., 2009; Shannon and O'tool, 2003). The two subsystems are influenced by two different source water masses, namely the South Atlantic Central Water (SACW) in the north and the Eastern South Atlantic Central Water (ESACW) in the south (McCartney, 1977; Shillington et al., 2006) which differ in their biogeochemical properties (Mohrholz et al., 2008; Flohr et al., 2014). Compared to the ESACW, the SACW is low in oxygen and enriched in dissolved nutrients, which distinguishes the OMZs in the two subsystems from each other. In the nBUS, the OMZ is essentially controlled by the seasonally variable inflow of the oxygen-poor SACW (Monteiro et al., 2006; Mohrholz et al., 2008), whereas the sBUS OMZ is assumed to be influenced to a much greater extent by the seasonally varying productivity and the resulting export production (Bailey, 1991; Pitcher and Probyn, 2011; Pitcher et al., 2014; Lamont et al., 2015). The consequences are that the sBUS OMZ develops predominantly in the bottom waters on the shelf while the nBUS OMZ also extends along the continental slope (Fig. 2). Although occasional mass mortality events of, e.g. rock lobsters, indicate that anoxic conditions occur in the shelf region (Cockcroft, 2001; Cockcroft et al., 2008; Hutchings et al., 2009), the oxygen concentrations in the sBUS OMZ are generally higher than in the nBUS OMZ. Accordingly, anoxic processes such as anammox and denitrification have a significant effect on the nitrogen cycle in the nBUS (Kalvelage et al., 2011; Nagel et al., 2013), and anoxic events, during which reduced gases such as $CH_4$ and $H_2S$ are released from the sediment, often occur in association with an increased influx of SACW in summer (Ohde and Dadou, 2018; Ohde et al., 2007).

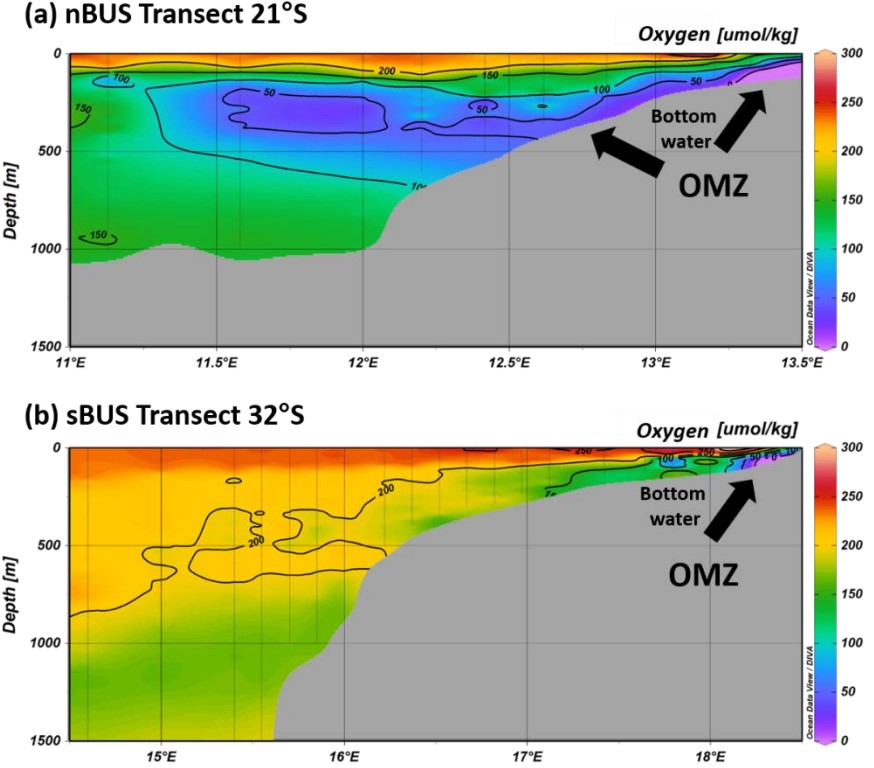

**Figure 2** Oxygen concentrations along a transect in the nBUS and sBUS during cruise SO285, showing the near-bottom OMZ in the nBUS (a) and sBUS (b), and the OMZ on the continental slope in the nBUS. Oxygen data (unpublished) from CTD casts conducted during cruise SO285. Graphic 1 and 2 created with Ocean Data View (Schlitzer, 2024).

Calculations of the upwelling velocities from wind fields in comparison with model results show that strong spatial variations occur within the two subsystems due to the interaction of wind stress and the geometry of the coast. Strong upwelling events at the coast can e.g., be accompanied by a weak upwelling in the adjacent ocean (Bordbar et al., 2021). In addition, mesoscale and sub-mesoscale processes such as eddies, filaments, the formation of oceanic fronts and vertical mixing influence the vertical water mass transport and with it the wind driven upwelling (Rixen et al., 2021b; Flynn et al., 2020; Bakun, 2017; Rubio et al., 2009). As a result, localised upwelling can exhibit a pronounced seasonality, such as off Walvis Bay in the nBUS, while the seasonality in other regions of the two subsystems is only weakly pronounced (Bordbar et al., 2021).

In contrast, sea surface temperatures (SST) in both subsystems show a pronounced seasonality with lower temperatures in winter and higher temperatures in summer (Fig. 3). Off Cape Columbine in the sBUS, winter cooling is accompanied by weaker upwelling while off Walvis Bay in the nBUS, stronger upwelling develops in phase with winter cooling whereas summer heating is associated with weaker upwelling (Bordbar et al., 2021).

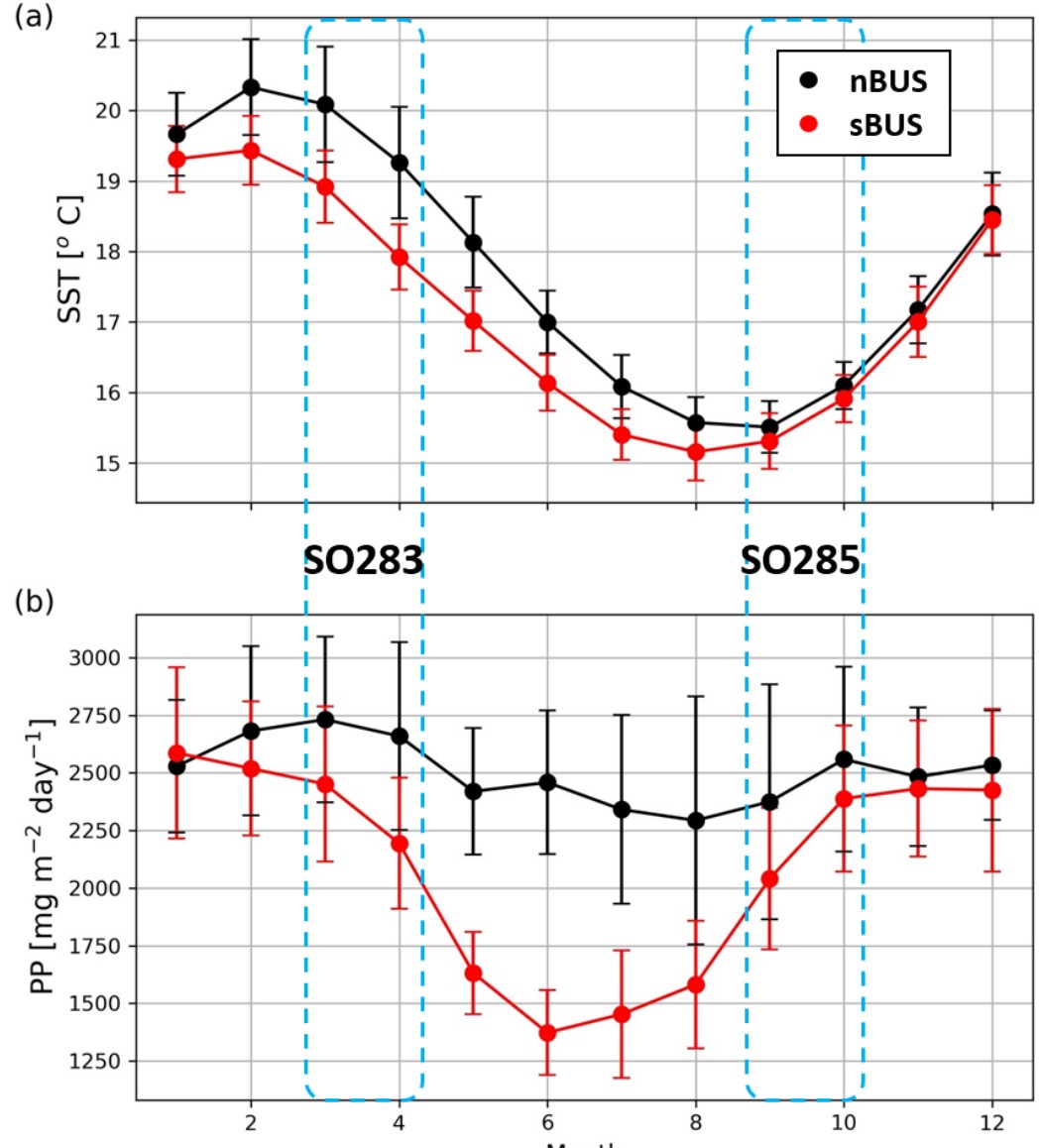

**Figure 3** Monthly mean annual cycles of sea surface temperatures (SST, a) and primary production rates (PP, b) in the nBUS and sBUS area, respectively. Satellite data (OI SST, and PP) have been downloaded in November 2023 (see method section). Dotted areas indicate the sampling periods of SO283 and SO285 cruises, respectively. Error bars indicate the standard deviation of PP and SST within the time period 1981-2023.

These results agree with observations along the Namibian monitoring line off Walvis Bay (23°S, Louw et al., 2016). They showed that, favoured by the weak upwelling and summer warming, a relatively strong stratification develops in the surface water at the beginning of the summer which dissipates at the end of the summer with the onset of winter cooling and the strengthening of the upwelling (Louw et al., 2016). Highest concentrations of chlorophyll were found in transitional phases, with clear maxima at the beginning and end of the summer between November and January, and March and April, respectively (Louw et al. 2016).

Averaged over the two subsystems, the primary production derived from satellite data (see Fig. 3) follows the seasonal pattern of chlorophyll concentration off Walvis Bay in both subsystems in so far as that primary production is lower on average in winter than in summer (average over 1981-2023). In the sBUS, however, the seasonality is much more pronounced than in the nBUS, which, as already mentioned, also affects the seasonality of the sBUS OMZ.

**Methods**

We deployed two long-term sediment trap systems to investigate the seasonality and the influence of zooplankton on the POC, one off Walvis Bay in the nBUS and one off Hondeklip Bay in the sBUS. In parallel, a total of 16 free-floating sediment trap arrays (drifters) were deployed on three research cruises (Table 3, Fig. 1). The moorings were equipped with Hydro-Bios MST 12 traps and for the drifter systems, Hydro-Bios Saarso single traps were used. Both trap versions have an identical cylinder design with a collection area of 0.015 m². The openings of the traps are fitted with a 2x2 cm honeycomb mesh to prevent macroparticles from entering and clogging the trap. The long-term moorings are part of ongoing long-term sediment trap studies in the BUS, considering for this study a mooring period between April 2010 and August 2022 off Walvis Bay for the nBUS (Rixen et al., 2021b; Vorrath et al., 2018) and between October 2019 and April 2022 in Hondeklip Bay for the sBUS (Rixen et al, 2021a) (Table 2, Fig. 6a). Both sites are located close to the coast line in water depths between 100 and 200 m. Hydro-Bios MST-12 sediment traps were moored at both stations in trap depths of approximately 64 and 100 m, respectively. The traps were equipped with twelve sample bottles (250 ml Nalgene) that rotated under the trap at fixed programmed intervals. The collection intervals of the individual sample bottles were between 9 and 40 days. The 16 drifters were deployed during the cruises with the German RVs Meteor (M153) and Sonne (SO283, SO285, Rixen et al., 2021b; Lahajnar et al., 2021). During the cruises a total of 83 single sediment traps were deployed along with the drifter arrays (Table 1). To keep the drifter in an upright position, they were equipped with a buoyancy unit at the upper end and a 30 kg ballast anchor at the lower end. The buoyancy unit also included an Iridium GPS transmitter in order to track the drifter during the deployment. Between the buoyancy unit and the ballast anchor, 4-7 Saarso single sediment traps were attached. The water depth at which sediment traps were deployed varied with bottom water depth. At water depths > 1000 m, the sediment traps were generally installed at water depths of 50 m, 100 m, 200 m, 300 m, 400 m and 500 m (see Table 3 for further details).

**Table 1** List of drifter-related cruises.

| Cruises | Start | End | Season | Ports of embarkation and arrival |
|---------|-------|-----|--------|----------------------------------|
| M153 | 2019-02-15 | 2019-03-31 | Summer 2019 | Walvis Bay, Namibia – Mindelo, Cape Verde |
| SO283 | 2021-03-19 | 2021-05-25 | Autumn 2021 | Emden – Emden, Germany |
| SO285 | 2021-08-20 | 2021-11-02 | Spring 2021 | Emden – Emden, Germany |

In order to prevent biological activity and degradation in the sample cups and to reduce exchange with the surrounding water, the water in the cups was poisoned with $HgCl_2$ (3.3 g/l) and enriched with salt (NaCl 70 g/l) before the deployment. In this way, the samples remain in the same condition as they were during the periods of deployment until the actual analysis: cooled, darkened and poisoned with $HgCl_2$. It has been accepted since the early 1980s that the addition of a toxin to sediment trap samples prevents bacterial or microbial degradation of the material (see e.g. Honjo et al., 1982). Metfies et al. (2017) have even found that PCR-based molecular genetic analysis is possible in sediment trap samples from long-term moorings when the samples have been poisoned with $HgCl_2$. After recovery, all samples were stored at 4°C on the ship and either examined directly on board or transported immediately to the home laboratory at the University of Hamburg, Germany, without interruption of cooling.

In general, all sediment trap samples were first macroscopically described and then divided into two fractions (> 1 mm and < 1 mm) using a 1x1 mm mesh size sieve. The > 1 mm fraction is classified as active swimmers (Lee

et al., 1988, 1991). The < 1 mm fraction represents the passive flux. Haake et al. (1993) and Rixen et al. (1996) described in detail the processing and analysis of sediment trap samples. Both the preparation of the sampling cup before deployment and the processing of the samples after recovery were carried out according to generally accepted procedures (e.g., Honjo et al., 1982 Honjo et al. 2008, Metfies et al., 2017). The > 1mm fraction of the long-term moorings will be discussed in a future work.


**Table 2** Overview of moored sediment trap deployments.

| Station | Mooring No | Latitude [°S] | Longitude [°E] | Water Depth [m] | Trap Depth [m] | Start | End | Sampling Interval [days] |
|---|---|---|---|---|---|---|---|---|
| nBUS | 1 | -23.0000 | 14.0800 | 140 | 70 | 2009-12-15 | 2010-08-26 | 21 |
| nBUS | 2 | -22.6000 | 14.2000 | 127 | 60 | 2010-11-27 | 2010-12-09 | 12 |
| nBUS | 3 | -23.0225 | 14.0277 | 130 | 75 | 2013-07-04 | 2013-07-31 | 30 |
| nBUS | 4 | -23.0248 | 14.0370 | 130 | 75 | 2014-05-14 | 2014-06-03 | 21 |
| nBUS | 6 | -23.0165 | 14.0368 | 130 | 75 | 2016-04-25 | 2016-05-24 | 29 |
| nBUS | 7 | -23.0173 | 14.0368 | 130 | 75 | 2017-04-26 | 2017-05-25 | 29 |
| nBUS | 8 | -23.0231 | 14.2185 | 110 | 65 | 2019-10-21 | 2019-11-30 | 40 |
| nBUS | 9 | -23.0229 | 14.0370 | 130 | 75 | 2021-07-01 | 2021-07-13 | 12 |
| nBUS | 10 | -23.0232 | 14.0370 | 130 | 75 | 2021-11-29 | 2021-12-08 | 9 |
| nBUS | 11 | -23.0233 | 14.0371 | 130 | 75 | 2022-07-22 | 2022-08-21 | 30 |
| sBUS | 1 | -30.6367 | 17.0158 | 170 | 95 | 2019-10-21 | 2019-11-20 | 30 |
| sBUS | 2 | -30.6374 | 17.0172 | 170 | 95 | 2021-07-02 | 2021-07-14 | 12 |
| sBUS | 3 | -30.6378 | 17.0175 | 170 | 95 | 2022-03-24 | 2022-04-23 | 30 |

**Table 3** Overview of drifting sediment trap deployments.

| Region | Cruise ID | Drifter | Water depth (m) | Deployment [UTC] | Recovery [UTC] | Dep. Lat [°S] | Dep. Lon [°E] | Rec. Lat [°S] | Rec. Lon [°E] | Drift Distance [km] | Number of Traps | Trap Depth [m] |
|---|---|---|---|---|---|---|---|---|---|---|---|---|
| nBUS | M153 | 1 | 2050 | 2019-03-03 22:36 | 2019-03-05 16:33 | 22.9996 | 12.2489 | 22.9358 | 12.1475 | 15.4 | 5 | 50, 100, 200, 300, 500 |
| nBUS | M153 | 2 | 1000 | 2019-03-06 22:18 | 2019-03-10 10:30 | 20.9999 | 11.9992 | 20.8942 | 11.9064 | 29.7 | 5 | 30, 50, 100, 150, 200 |
| sBUS | M153 | 3 | 1300 | 2019-02-22 18:51 | 2019-02-25 12:13 | 31.0440 | 15.2280 | 30.9683 | 15.095 | 20.5 | 5 | 50, 100, 200, 300, 500 |
| sBUS | M153 | 4 | 160 | 2019-02-18 11:07 | 2019-02-20 08:27 | 30.6417 | 17.0226 | 30.5275 | 16.8453 | 24.4 | 4 | 20, 30, 50, 75 |
| sBUS | SO283 | 5 | 997 | 2021-04-15 19:58 | 2021-04-17 12:33 | 31.4987 | 15.4980 | 31.4802 | 15.4353 | 8.5 | 6 | 50, 100, 200, 300, 400, 500 |
| sBUS | SO283 | 6 | 192 | 2021-04-16 09:15 | 2021-04-18 05:52 | 30.9998 | 16.9998 | 31.0366 | 16.9735 | 11.4 | 5 | 20, 30, 50, 75, 100 |
| nBUS | SO283 | 7 | 1155 | 2021-04-21 09:16 | 2021-04-23 14:26 | 23.0001 | 12.7496 | 22.9750 | 12.6905 | 9.4 | 5 | 50, 100, 300, 400, 500 |
| nBUS | SO283 | 8 | 248 | 2021-04-21 05:49 | 2021-04-23 10:19 | 23.0001 | 13.5833 | 22.9887 | 13.4818 | 15.6 | 5 | 20, 30, 50, 75, 100 |
| nBUS | SO283 | 9 | 732 | 2021-04-24 14:41 | 2021-04-25 15:07 | 17.9998 | 11.3001 | 17.9641 | 11.3210 | 5.6 | 7 | 10, 50, 100, 200, 300, 400, 500 |
| nBUS | SO283 | 10 | 360 | 2021-04-27 10:57 | 2021-04-29 11:19 | 25.0004 | 13.9151 | 24.868 | 13.6998 | 22.1 | 5 | 20, 30, 50, 75, 100 |
| nBUS | SO283 | 11 | 1045 | 2021-04-27 04:59 | 2021-04-29 06:00 | 25.0002 | 13.3333 | 24.8684 | 13.2678 | 18.1 | 4 | 50, 100, 300, 400 |
| sBUS | SO285 | 12 | 1097 | 2021-09-19 02:50 | 2021-09-19 15:48 | 31.9298 | 15.7738 | 31.9298 | 15.7738 | 1.91 | 5 | 20, 50, 100, 150, 200 |
| sBUS | SO285 | 13 | 155 | 2021-09-21 01:22 | 2021-09-21 14:44 | 30.9962 | 17.3493 | 30.9180 | 17.3133 | 9.95 | 5 | 20, 30, 50, 75, 100 |
| sBUS | SO285 | 14 | 1047 | 2021-09-24 07:57 | 2021-09-25 14:15 | 30.3525 | 14.5834 | 30.3296 | 14.4208 | 17.8 | 6 | 50, 100, 200, 300, 400.0, 500 |
| nBUS | SO285 | 15 | 1888 | 2021-10-01 19:35 | 2021-10-02 15:35 | 22.9981 | 12.3988 | 22.8982 | 12.4168 | 18.01 | 6 | 50, 100, 200, 300, 400, 500 |
| nBUS | SO285 | 16 | 517 | 2021-10-05 07:19 | 2021-10-07 12:01 | 21.0223 | 12.4218 | 20.7052 | 12.4466 | 35.97 | 5 | 20, 50, 100, 150, 200 |


For the drifter samples, the fraction > 1 mm was analysed for active simmers by the use of Keyence VHX-6000 digital microscope. It should be noted that there are currently no definitive, generally standardised methods for quantifying the proportion of migrating zooplankton that has actively entered the sediment trap. However, methods such as that of Weldrick et al. (2021), where the active organisms were hand-picked to quantify their abundance

in drifting sediment trap samples, provide approaches dealing with active swimmers in drifting sediment trap samples; this procedure is similar and comparable to the approach used in this study. After removing swimmers from the > 1mm fraction, they were classified according to Tutasi and Escribano (2020), Ekau et al. (2018) and Castellani and Edwards (2017). Furthermore, zooplankton > 1 mm without any signs of decay or disintegration were classified as swimmers and included to the active flux. Zooplankton that showed clear signs of degradation

or disintegration were considered part of the passive flux, as we assumed these organisms were dead when they entered the trap.

The active flux (i.e., swimmers) were divided into eight groups: amphipoda, copepoda, decapoda, euphausiacea, ostracoda, pteropoda, fish larvae and gelatinous zooplankton. Swimmers which could not be further classified were therefore classified as 'zooplankton unknown'.

Subsequently, all trap samples used for further analyses were taken on polycarbonate filters (Millipore, 0.45 μm mesh size). The samples were rinsed with a sodium tetraborate buffer solution (2 mg $Na_2[B_4O_5(OH)_4]$ $8H_2O$ per 1 litre $H_2O$) to remove salt and prevent dissolution of carbonates. After filtration, the filters were dried at 40°C for 48 hours and weighed to determine the dry weight in each zooplankton group and of the < 1 mm fraction. The dried material was carefully removed from the filters with a spatula in order to prevent contamination of the sample

by filter material. All filters were thoroughly visually inspected to ensure that no sample material larger than 0.45 μm (mesh size of the polycarbonate filter) remained on the filter surface. Virtually blank filters were left behind, thus preventing size fractionation. The material was then homogenised using an agate mortar and pestle and subsequently analysed for total carbon (TC) and total nitrogen (TN). POC was measured in a second run of samples in which inorganic carbon was removed by acidification (1N HCl). All analyses were carried out using

the flash combustion method (Euro Vector EA-3000). Due to the lack of sufficient sample material, e.g., when analysing swimmers, it was often only possible to determine the TC. As swimmers contain almost no carbonate apart from pteropods, the measured TC was interpreted as POC. This leads to a potential overestimation of the active POC flux, but given the relatively low abundance of this zooplankton clade, as shown in the results section, we expect this error to be negligible. The active and the passive POC flux together give the total POC flux. The

POC flux multiplied by 1.8 (Anderson, 1995, Francois et al., 2002) results in the organic matter (OM) flux.

Satellite-derived monthly mean sea surface temperature (SST, Reynolds et al., 2002) and net primary production rates (Behrenfeld and Falkowski, 1997) were downloaded from the OI-SST website (http://iridl.ldeo.columbia.edu/SOURCES/.NOAA/.NCEP/.EMC/.CMB/.GLOBAL/.Reyn_SmithOIv2/.monthly/.sst/) and the Ocean Primary Production website (http://www.science.oregonstate.edu/ocean.productivity/) in

November 2023. The SST data with a spatial resolution of 0.33x0.33 degrees covered the period from 1981 to 2023, while the primary production rates with a resolution of 1x1 degrees covered the periods from 2002 to 2023. When calculating the mean values for the BUS and the two subsystems, it was assumed that the BUS covers the area between the coast and about 250 km offshore, within the latitudes we have previously specified (nBUS 17-27°S and sBUS 27°-35°S, see Fig.1).

## Results

During the research cruises, the primary production rates at the coast reached values of $> 9000$ mg m$^{-2}$ day$^{-1}$, which decreased with increasing distance from the coast and fell to values of $< 10$ mg m$^{-2}$ day$^{-1}$ far offshore (Fig. 4). The primary production rates at the drifter positions varied between 544.6 mg m$^{-2}$ day$^{-1}$ and 5115.4 mg m$^{-2}$ day$^{-1}$ and revealed a mean of $1618.6 \pm 1110.6$ mg m$^{-2}$ day$^{-1}$. They thus fell below the average primary production rates, which were 2505.3 mg m$^{-2}$ day$^{-1}$ in the nBUS and 2089.6 mg m$^{-2}$ day$^{-1}$ in the sBUS (Fig. 3b). Overall, the primary production derived from the satellite data largely fell within the range of primary production rates determined during research cruises in winter 1999 and summer 2002 in the sBUS and nBUS ($140 - 8830$ mg m$^{-2}$ day$^{-1}$ Barlow et al., 2009).

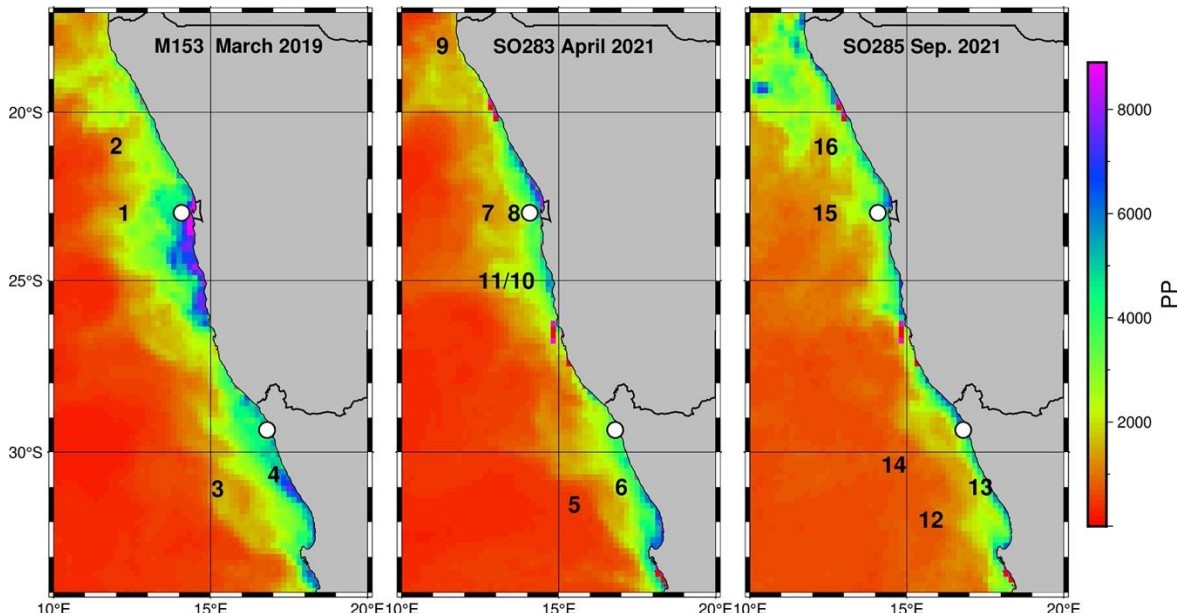

**Figure 4** Primary production rates (SeaWiFS) in mg C m$^{-2}$ day$^{-1}$ during the research cruises M153, SO283 and SO285 as well as the locations of the drifter deployments (black numbers) and long-term sediment trap mooring sites (white circles).

The results from the 83 drifter traps showed that copepod biomass accounts for the largest proportion of the active POC flux in both subsystems averaged over all water depths (55.7 % and 46.2 % in the nBUS and sBUS), followed by the biomass of amphipods and euphausiids (Fig. 5). The combined proportion of the three groups (copepods, amphipods and euphausiids) amount to about 77.3 % and 87.4 % of the active POC in the nBUS and sBUS.

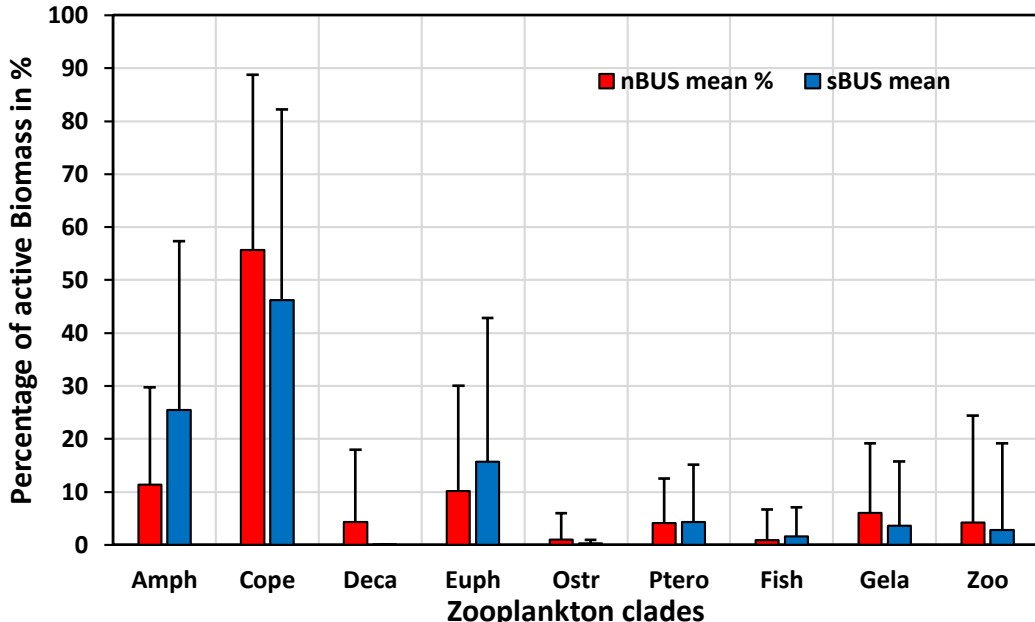

**Figure 5** Proportion of zooplankton groups in the active POC averaged over all traps with a mean standard deviation of 15±8%. Amph- amphipods, Cope- copepods, Deca – decapoda, Euph- euphausiids, Ostr- ostracodes, Ptero- pteropods, Fish- fish larvae, Gela – gelatinous organisms, Zoo- zooplankton not further identified.

Averaged over all sampled water depths and drifter deployments in both subsystems the proportion of the active POC flux to the total POC flux was on average 10.9% higher in the sBUS (mean: 72.9 %) than in the nBUS (mean: 62.0 %) and varied with depth by ± 14.2 % (nBUS) and ± 9.0 % (sBUS). However, the active POC flux of 944.5 ± 743.6 mg m$^{-2}$ day$^{-1}$ in the sBUS was 2.9 times higher than in the nBUS with 322 ± 231.7 mg m$^{-2}$ day$^{-1}$. This difference was particularly visible in the upper water column (water depths < 100 m), and decreased in greater water depths where the active POC flux in nBUS could even exceed that in the sBUS (Fig. 7a).

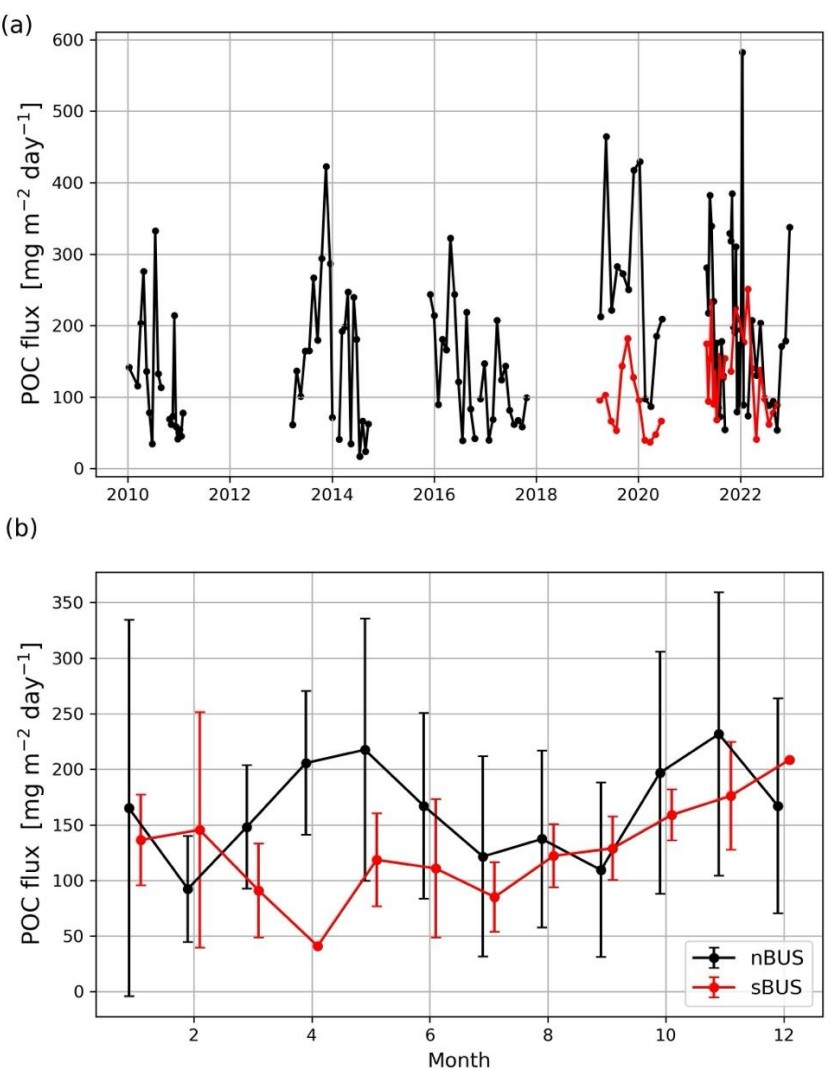

**Figure 6** POC flux rates measured off Walvis Bay in the nBUS and Hondeklip Bay in the sBUS moorings (a) and the mean annual cycle (b) derived from the mooring data shown in (a).

Compared to the active POC flux, the passive POC flux in the sBUS is only slightly higher (293.9 $\pm$ 249.0 mg m$^{-2}$ day$^{-1}$) than in the nBUS (203.1 $\pm$ 157.0 mg m$^{-2}$ day$^{-1}$). The passive POC flux tends to decrease with depth (Fig. 7b) and shows an average POC flux of 135.9 $\pm$ 82.3 mg m$^{-2}$ day$^{-1}$ (nBUS) and 365.1 $\pm$187.3 mg m$^{-2}$ day$^{-1}$ (sBUS) at a water depth of 100 m. In contrast, the POC flux in the moored traps averaged over the entire observation period in the nBUS at a water depth of 64 m (moored trap depth) and in the sBUS at a water depth of

about 100 m (moored trap depth) was 169.8 $\pm$ 128.6 mg m$^{-2}$ day$^{-1}$ and 120.1 $\pm$ 55.8 mg m$^{-2}$ day$^{-1}$, respectively. The data from the moored sediment traps showed no clear seasonal variability, but in the nBUS there was significant interannual variability (Fig. 6). However, our time series in the sBUS is still too short to make statements about interannual variability.

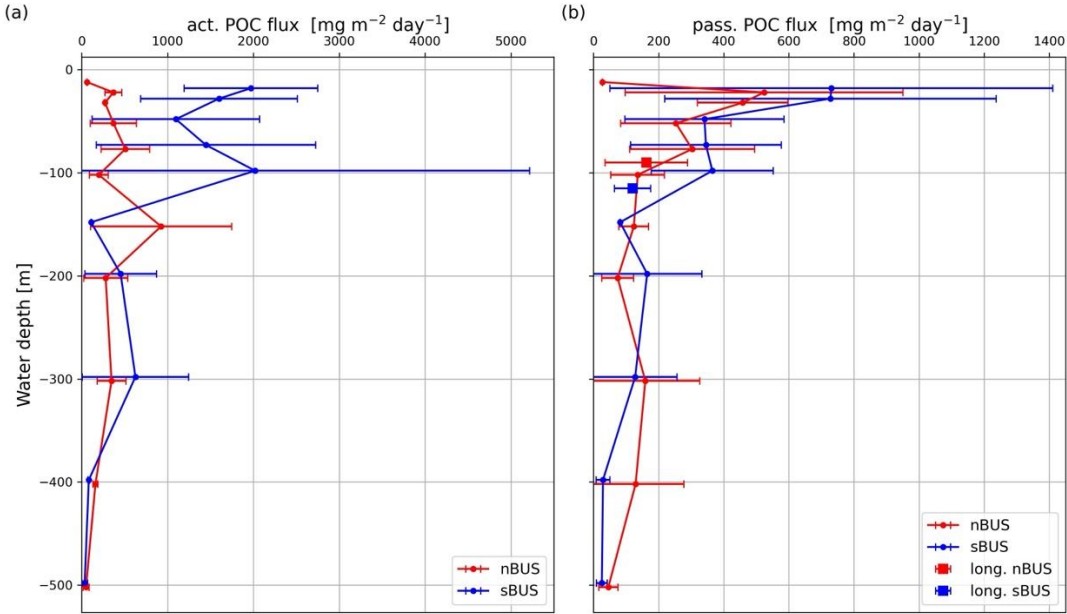

**Figure 7** Active (a) and passive POC flux (b) averaged over all traps deployed at the same water depth versus water depth, including mean passive POC flux rates (squares) of the long-term moorings in the nBUS and sBUS.

## Discussion

The drifters were deployed on the shelf and along the continental slope at the beginning and end of the summer season. However, as mentioned before, the drifters were deployed in areas where the primary production was on
average below the mean summer primary production and it ranged between 544.6 and 5114.4 mg m$^{-2}$ day$^{-1}$. The active and passive POC flux in the upper water column (water-depth < 100 m) varied between 45.3 and 9121.2 mg m$^{-2}$ day$^{-1}$ and 8.8 – 1878 mg m$^{-2}$ day$^{-1}$ respectively. This means that the variability of the active POC flux exceeded that of primary production, and the variability of the passive POC flux, which is comparatively low, still extends over three orders of magnitude (Fig. 7b). These large variations are the product of the interaction
between the oceanographic processes that influence the transport of nutrients into the euphotic zone, their horizontal distribution in the surface water and biological processes that convert the nutrients into biomass and export them as POC. Our data density is not sufficient to disentangle this complex interaction completely, but on average the sediment trap results agree well with the results of other studies. For instance, vertical hauls down to a water depth of 600 m with a Hydro-Bios Multi Plankton Sampler in the Humboldt Upwelling System off northern
Chile yielded a mean biomass of migrating zooplankton of 958 mg C m$^{-2}$ day$^{-1}$ (Tutasi and Escribano, 2020), which is quite similar to the mean active POC flux derived from our drifter trap samples (944.5 ± 743.6 mg m$^{-2}$ day$^{-1}$ in the sBUS and 322 ±- 231.7 mg m$^{-2}$ day$^{-1}$ in nBUS). In the California Upwelling System, the active and passive POC flux at a water depth of 100 m was estimated at 34.8 and 108.0 mg m$^{-2}$ day$^{-1}$, respectively (Stukel et al., 2023). This active POC flux is below the mean flux we have measured in nBUS at a water-depth of 100 m (200
± 108 mg m$^{-2}$ day$^{-1}$) but it falls in the range of the active flux we have determined in the sBUS (2022.1 ± 3195 mg m$^{-2}$ day$^{-1}$) at a water-depth of 100 m (Fig. 7a). The passive flux off California, on the other hand, is in the range of what we have measured with the drifter in the nBUS (135.8 ± 82 mg m$^{-2}$ day$^{-1}$, Fig. 7b) but falls below what was measured in the sBUS (365.1 ±187.2 mg m$^{-2}$ day$^{-1}$, Fig. 7b). Overall, our results are in good agreement with those from other eastern boundary upwelling systems.

Our sediment trap results show furthermore that copepods dominate the active POC flux in both subsystems (Fig. 5), which is consistent with biological studies revealing that copepods dominate the abundance of mesozooplankton in the nBUS and sBUS (Verheye et al., 2016; Bode et al., 2014). This implies that copepods in the BUS not only play a key role as a food source for the conservation of marine fish stocks, but are also of great importance for the active POC flux. This in turn raises the question of how the active flux affects the passive POC flux and the associated POC transport onto the sediments.

Zooplankton influence the passive POC flux to the sediment in many ways, which can increase or decrease the active POC flux (Steinberg and Landry, 2017, Boyd et al., 2019, Moigne 2019 and Miles, 2018). Grazing and disaggregation are e.g., processes that decrease the passive POC flux while excretion of faecal pellets and the death of zooplankton increase the passive POC flux (Cavan et al., 2020, Turner, 2014, Ducklow et al., 2001). In the sBUS, the active POC flux was almost 3 times higher than in the nBUS, which was most pronounced in the upper water column (water depth < 150 m, Fig 7 a), while the passive POC flux did not differ greatly between the two subsystems (Fig. 7b). This implies that the role of zooplankton on the passive POC flux varied in the two subsystems.

On average, the passive POC flux decreases with depth as commonly observed in other studies and has been described by a POC flux attenuation equation (Cael and Bisson, 2018; Martin et al., 1987; Giering et al., 2014). To ensure comparability with other regions, we have selected the commonly used equation (Eq. 1, Martin et al. 1987) for describing the POC flux attenuation at increasing water-depth and adapt it to the situation found in the BUS (Fig. 8).

The selected MLD as well as the determined $F_{MLD}$ and 'b' values were then used to calculate the POC fluxes using the Martin curve and the water depth. The calculated and measured POC fluxes correlated with each other ($r = 0.925$, n = 31) and showed the best agreement at an assumed MLD of approximately 10 m. The values for $F_{MLD}$ and 'b' obtained from the curve fitting were 1117 mg C m$^{-2}$ day$^{-1}$ and -0.74 (Fig. 8). Temperature profiles obtained during our expeditions at our long-term sediment trap sites showed that MLDs in the nBUS and sBUS varied between approximately 30 m and 15 m during the austral spring (Fig. 9a). In late summer, the mean depth varied between about 1 m and 14 m (Fig. 9b), showing that an MLD of 10 m, as assumed for the curve fitting, is within the range observed during our expeditions.

If the 'b' is reduced to -0.86, the resulting POC flux attenuation curve represents the long-term sediment trap data off Walvis Bay in nBUS and Hondeklip Bay in sBUS significantly better. However, 'b' values of -0.74 and -0.86 are in the range of the 'b' values found in the Pacific Ocean and the Atlantic Ocean (b = -0.5 to -1.38; Giering et al., 2014) and similar to those determined in the California upwelling system (b = -0.72, Stukel et al., 2023). The calculated POC flux rates also agree well with those measured in previous sediment trap studies in the BUS at water depth > 500 m (Fig. 8a, Vorrath et al., 2018).

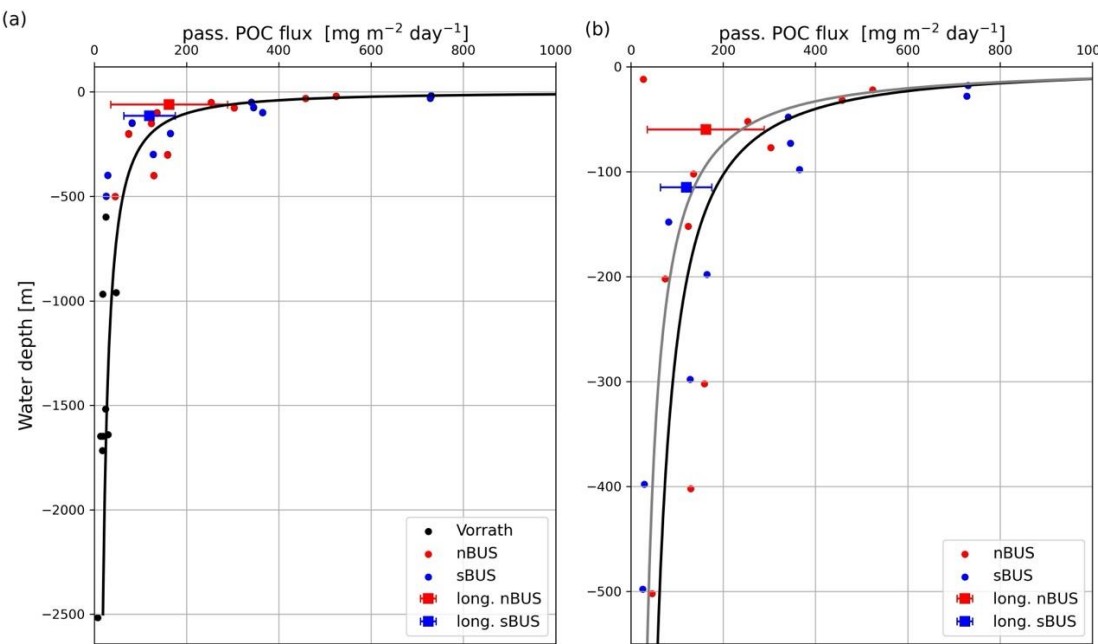

345 **Figure 8** Mean POC flux rates (red and blue dots) versus water depths of 0 m - 2500 m (a) and 0 m - 500 m (b). The curves are Martin curves with a 'b' of -0.74 (black) and -0.86 (grey). The result of an nBUS trap at 10 m water depth (see b, red circle at 10 m depth) was not considered in the curve fitting. Sediment trap data (black dots) taken from Vorrath et al. (2018); squares depict the mean POC flux of the long-term moorings from this study.

The determined $F_{MLD}$ implies a mean export production of 1117 mg C m$^{-2}$ day$^{-1}$. Compared to the primary

350 production rates of 2089 mg C m$^{-2}$ day$^{-1}$ (sBUS) and 2505 mg C m$^{-2}$ day$^{-1}$ (nBUS) this indicates an f-ratio (=export production/primary production) of approximately 0.4 to 0.5 which is characteristic of highly productive systems (Eppley and Peterson, 1979). This means that the passive POC flux determined with drifters and long-term sediment traps (on the shelf and in the open ocean) appears to follow the general attenuation equations on average, despite the large variations of the individual measurements.

355

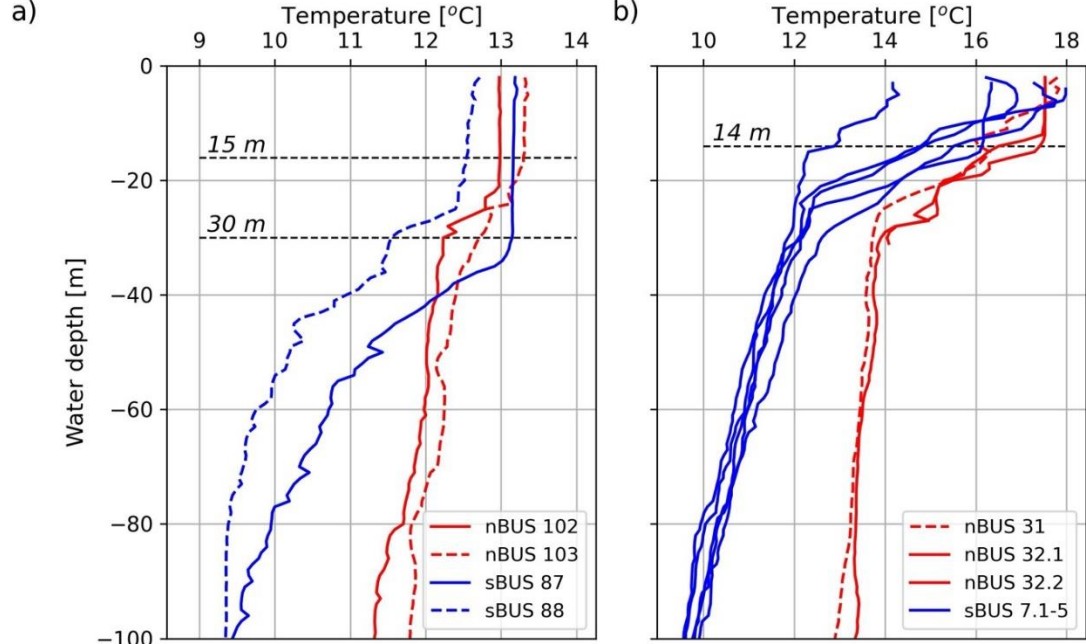

**Figure 9** Temperature profiles at the CTD stations (depth resolution of 1 m) closest to the long-term mooring site during the cruises SO285 austral spring (a) and M153 austral summer (b). The positions of the mooring stations are shown in Fig. 1 and Tab.1. Decimal places indicate that there have been several casts at the stations. Station 7 was a long station at which five CTD casts have been carried out over the course of a day. It should be noted that the MLD at station 7 showed a diurnal variation, with a maximum depth of approx. 14 meters. A maximum MLD of approx. 14 m was also shown in the nBUS (station 32). During the SO285 cruise in austral spring, the MLDs varied between approx. 30 m and 15 m. The dashed lines were inserted to illustrate the maximum MLDs. CTD data available at https://www.pangaea.de.

Assuming a mean water depth of 150 m, which is based on the mean depth of the continental shelf, the derived Martin equations (b of -0.74 and -0.86, respectively) suggest a POC supply to sediments of 108.8 - 152.0 mg C m$^{-2}$ day$^{-1}$. Compared to the primary production rate this implies that 4 – 7 % of primary produced POC reaches the sediment. This is consistent with our former result of 5 % based on the long-term sediment trap study off Walvis Bay and contradicts the results of a numerical model, which states that 49 % of the primary produced POC reaches the sediment surface (Emeis et al., 2018). However, the similar POC fluxes in nBUS and sBUS are remarkable, especially considering the differences in the active POC flux and the OMZ intensity (Fig.10).

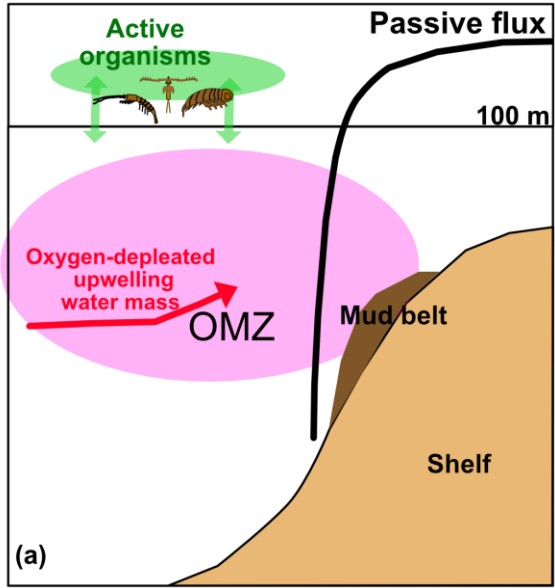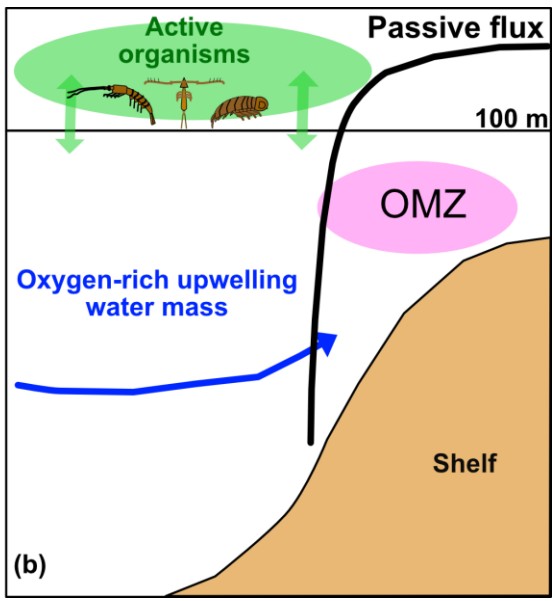

**Figure 10** System schematics in the nBUS (a) and the sBUS (b). Green arrows and their size characterize the active transport pathways. The black curve shows the Martin curve based passive POC flux which is similar in both systems. Red and blue arrows show the contribution and entering of upwelling source water masses. The dark brown area indicates the POC-rich mud belt in the nBUS which is absent in the sBUS.

Low oxygen concentrations and the associated transition from oxic to anoxic degradation processes are considered to be a factor that reduces POC degradation in the water column and surface sediments (Paropkari et al., 1993; Laufkötter et al., 2017) and also increase the quality of the preserved organic matter in the nBUS (Nagel et al., 2016). If lower oxygen concentrations in the nBUS have a positive effect on the POC flux, but this flux, including export production, is almost similar to that in the sBUS, there are probably also processes that, similar to the low oxygen concentration in the nBUS, favour the passive POC flux in the sBUS. Since the active POC flux in the upper water column is significantly higher in the sBUS than in the nBUS (Fig. 7a), zooplankton is assumed to play this role. As primary production and export production rates are similar in both subsystems, high zooplankton abundance is expected to favour passive POC fluxes by enhancing the production of fast-sinking faecal pellets in the sBUS. As mentioned above, the attenuation coefficient 'b' is an expression of the strength of remineralisation. According to our results, 'b' is similar in both subsystems, which leads to the assumption that the effect of a lower oxygen concentration on 'b' in the nBUS is comparable to the effect of zooplankton on 'b' in the sBUS. One way in which zooplankton may contribute is by increasing the formation of rapidly sinking faecal pellets and reducing the residence time of sinking particles in the water column. This would mean that the positive effect on passive POC flux caused by higher zooplankton abundance would overcompensate for negative effects such as increased disaggregation of sinking particles due to grazing at our study sites.

In the water column, however, the effects of POC degradation processes on the distribution of nutrients and dissolved inorganic carbon are masked by the inflow of different upwelling source waters in the two subsystems as illustrated in Figure 10. As mentioned above, and shown by Siddiqui et al. (2023), source waters in the nBUS are not only oxygen depleted and nutrient enriched but also enriched in dissolved inorganic carbon (DIC). The much more pronounced OMZ in the nBUS, in addition to the almost equal POC fluxes in both subsystems and the high sedimentary POC concentrations in the nBUS (Fig. 1), also indicates that the mud belt in the nBUS is largely a consequence of the stronger OMZ in the nBUS compared to the sBUS. This highlights the ambivalent nature of

expanding OMZs, which mitigates atmospheric and oceanic $CO_2$ accumulation by increasing POC storage in sediments and poses a threat to established ecosystems and fisheries.

**Summary**

Our results indicate that the large variability of the measured POC fluxes reflects the expected spatial and temporal variability of the BUS. On average, however, and in agreement with other studies, our results show that copepods

dominate zooplankton abundance and the active POC flux in both subsystems. The active POC flux in the sBUS was almost 3 times higher than the nBUS. This difference was particularly evident in the upper 100 to 150 m of the water column. In contrast to the active POC flux, the passive POC flux was almost the same in both subsystems and followed in accordance with the long-term sediment trap data the general attenuation equations on average, despite significant deviation of the individual measurements. A similar POC export production and attenuation

with increasing water depth, in addition to a more pronounced OMZ in the nBUS and a much higher zooplankton abundance in the sBUS implies that both, the low oxygen concentrations and the higher zooplankton abundance, favour the passive POC flux at our study sites. The former is well accepted while the latter means that zooplankton reduce POC remineralization by increasing the particle sinking speed. This, in turn, suggests that increased formation of fast-sinking faecal pellets overcompensates for enhanced grazing on sinking particles that favours

particle disaggregation and reduces the sinking speed. Similar productivity and POC supply but large differences in POC concentration and storage in the surface sediments of the nBUS and sBUS suggest that the intensity of the OMZ is key to understanding the development of the POC-rich mud belt in the nBUS which is absent in the sBUS.

*Author contributions.* Conception and design of study: LM, NL, TR; acquisition of data: NL, LM, TL, analysis and interpretation of data: LM, TR, NL; Drafting the manuscript: TR, LM; revising the manuscript critically for important intellectual content: NL, AvdP, TL.

*Competing interests.* The contact author has declared that none of the authors has any competing interests.


ther geographical representation in this paper. While Copernicus Publications makes every effort to include appropriate place names, the final responsibility lies with the authors.


*Acknowledgements.* We would like to thank all scientists, technicians, captains and crew members of the German and South African research vessels Meteor, Sonne and Algoa for their support during the cruises M153, ALG 269, SO283 and SO285. In particular, we are very grateful to the DFFE mooring led by Tarron Lamont and Marcel van den Berg, for successful recovery and new deployment of the Hondeklip Bay mooring during the RS Algoa cruises

ALG 269 and ALG 285 in 2020 and 2022. Furthermore, we want to thank Marc Metzke and Frauke Langenberg for their excellent technical support during lab analyses with the samples at the Universität Hamburg.

*Financial support.* This research was funded by the German Federal Ministry of Education and Research (BMBF) under the grant no. 03F0797A (ZMT) and 03F0797C (Universität Hamburg).


The article processing charges for this open-access publication were covered by ZMT Leibniz-Centre for Tropical Marine Research, Bremen.

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

**Figure 1** POC concentration of surface sediment samples in the BUS from various research expeditions published in Emeis et al. (2018). Yellow stars show deployment drifter positions; red diamonds show long-term mooring locations. Contour lines in 1.5 % steps. Red dashed areas indicate the nBUS and sBUS, respectively (Hutchings et al., 2009).

**Figure 2** Oxygen concentrations along a transect in the nBUS and sBUS during cruise SO285, showing the near-
bottom OMZ in the nBUS (a) and sBUS (b), and the OMZ on the continental slope in the nBUS. Oxygen data (unpublished) from CTD casts conducted during cruise SO285. Graphic created with Ocean Data View (Schlitzer, 2024).

**Figure 3** Monthly mean annual cycles of sea surface temperatures (SST, a) and primary production rates (PP, b) in the nBUS and sBUS area, respectively. Satellite data (OI SST, and PP) have been downloaded in November
2023 (see method section). Dotted areas indicate the sampling periods of SO283 and SO285 cruises, respectively. Error bars indicate the standard deviation of PP and SST within the time period 1981-2023.

**Figure 4** Primary production rates (SeaWiFS) in mg C m$^{-2}$ day$^{-1}$ during the research cruises M153, SO283 and SO285 as well as the locations of the drifter deployments (black numbers) and long-term sediment trap mooring sites (white circles).

**Figure 5** Proportion of zooplankton groups in the active POC averaged over all traps with a mean standard deviation of 15±8%. Amph- amphipods, Cope- copepods, Deca – decapoda, Euph- euphausiids, Ostr- ostracodes, Ptero- pteropods, Fish- fish larvae, Gela – gelatinous organisms, Zoo- zooplankton not further identified.

**Figure 6** POC flux rates measured off Walvis Bay in the nBUS and Hondeklip Bay in the sBUS moorings (a) and the mean annual cycle (b) derived from the mooring data shown in (a).

**Figure 7** Active (a) and passive POC flux (b) averaged over all traps deployed at the same water depth versus water depth, including mean passive POC flux rates (squares) of the long-term moorings in the nBUS and sBUS.

**Figure 8** Mean POC flux rates (red and blue dots) versus water depths of 0 m - 2500 m (a) and 0 m - 500 m (b). The curves are Martin curves with a 'b' of -0.74 (black) and -0.86 (grey). The result of an nBUS trap at 10 m water depth (see b, red circle at 10 m depth) was not considered in the curve fitting. Sediment trap data (black dots) taken
from Vorrath et al. (2018); squares depict the mean POC flux of the long-term moorings from this study.

**Figure 9** Temperature profiles at the CTD stations (depth resolution of 1 m) closest to the long-term mooring site during the cruises SO285 austral spring (a) and M153 austral summer (b). The positions of the mooring stations are shown in Fig. 1 and Tab.1. Decimal places indicate that there have been several casts at the stations. Station 7 was a long station at which five CTD casts have been carried out over the course of a day. It should be noted that
the MLD at station 7 showed a diurnal variation, with a maximum depth of approx. 14 meters. A maximum MLD of approx. 14 m was also shown in the nBUS (station 32). During the SO285 cruise in austral spring, the MLDs varied between approx. 30 m and 15 m. The dashed lines were inserted to illustrate the maximum MLDs. CTD data available at https://www.pangaea.de.

**Figure 10** System schematics in the nBUS (a) and the sBUS (b). Green arrows and their size characterize the
active transport pathways. The black curve shows the Martin curve based passive POC flux which is similar in both systems. Red and blue arrows show the contribution and entering of upwelling source water masses. The dark brown area indicates the POC-rich mud belt in the nBUS which is absent in the sBUS.

**Table Caption**

**Table 1** List of drifter-related cruises.
**Table 2** Overview of moored sediment trap deployments.

**Table 3** Overview of drifter deployments.