# Peer review of "The influence of zooplankton and oxygen on the particulate organic carbon flux in the Benguela Upwelling System"

_EGUsphere, 2024_

## Author Comment (AC2)

[Figure]

**Fig. 1** Map showing the two working areas: nBus and sBus, as well as CTD stations near the long-term mooring sites. The positions of the long-term mooring sites are marked with open circles. The CTD stations of the cruise M153 and SO285 are marked with blue and red circles. The numbers are the station numbers.

[Figure]

**Fig. 2** Temperature profiles at the CTD stations (depth resolution of 1 m) closest to the long-term mooring site during the cruises SO285 (a) and M153 (b). The positions of the stations are shown in Figure 1. Decimal places indicate that there have been several casts at the

stations. Station 7 was a long station at which five CTD casts have been carried out over the course of a day. It should be noted that the MLD showed a diurnal variation, with a maximum depth of approx. 14 meters. A maximum MLD of approx. 14 m was also shown in the nBUS (station 32). During the SO285 cruise in the southern winter, the MLDs varied between approx. 30 m and 15 m. The dashed lines were inserted to illustrate the maximum MLDs.

[Figure]

**Fig. 3** The preliminary system schematics left nBUS and right sBUS, with all the relevant processes described. the black curve here indicates the passive flux. OMZ Oxygen Minimum Zone

---

## Author Response (AR1)

**Biogeosciences: egusphere-2024-700**

The influence of zooplankton and oxygen on the particulate organic carbon flux in the Benguela Upwelling System

Luisa Chiara Meiritz, Tim Rixen, Anja Karin van der Plas, Tarron Lamont, and Niko Lahajnar

**Reviews and Comments**

07. August 2024

We are indebted to the editor and to the reviewers for their valuable comments and suggestions for the improvement of our study. We have followed up all comments in detail and have revised the manuscript in many areas, including a fundamentally new Methods section. All changes are marked in blue in the current version to make it easier to follow the differences to the original. We have addressed the individual comments of the reviewers as described below.

**Referee #1**

*We would like to thank the anonymous reviewer for his valuable comments and suggestions for corrections. We would like to address the specific points in detail below.*

I have some concerns regarding the methodology. The > 1 mm particle fraction is analysed for zooplankton group identification and deduction of biomass flux. To what extent are these composed of sinking zooplankton and swimmers who got caught accidentally on the traps? Authors just briefly mention this possibility but do not enter into a detailed discussion about this issue

- *We agree that it is indeed difficult to distinguish whether the zooplankton organisms and groups were still alive when they entered the sediment trap or whether they were already dead organisms. Sediment trap experiments have been carried out since approximately 1978 (Smetacek et al., 1978) and since than this problem is known (Lee et al., 1991). It was not solved but there is general agreement to a distinct between sinking particles <1 mm and >1 mm. The fraction <1 mm is considered as passive (true) particle flux, the fraction >1 mm as active "swimmers" (Honjo et al., 2008). This allows to compare particle flux studies conducted worldwide, which we have also referred to in this manuscript. (see revised Method section)*
- *We now went one step further and examined the organisms in detail. We were able to distinguish whether the organisms were still intact or already showing signs of degradation and disintegration. Judging by the assumption that organisms in the poisoned $HgCl_2$ solution die in a relatively short time, we assume that their condition in the trap corresponds to that with which the organisms entered the trap. Hence, each organism was analyzed microscopically for traces of biological and physical degradation processes and sorted optically. Organisms that showed obvious traces of degradation were considered as passive flux as we assumed that they were dead as they entered the trap. In case they were no signs of degradation we counted the organisms as swimmers. (see revised Method section)*

When determining biomass weight and elemental composition, carbonate content is not taken into account (as mentioned by the authors), although pteropods appear to be present. What impact would pteropod carbonate have?

- *Yes, it is true, carbonate shells naturally also contain carbon, so do pteropods. However, in our study we have refrained from addressing the topic of the carbonate system and the uptake, conversion and release of carbon as carbonate carbon in order not to lose sight of the focus of the work. Whenever there was sufficient material available, PIC contents were also analyzed and will also be reported in the methods section an error analysis for POC and PIC data will also be added, to clarify our outcomes. (see lines 230-233)*

What is the time lag between sampling and laboratory analysis? Is the used protocol safe for ensuring biological material remains unaltered till analysis?

- *After recovery, the samples are kept cool on the ship and either directly analyzed on board or after the end of the expeditions transported by refrigerated air freight to the home laboratories immediately. The samples therefore remain in the same state as during the mooring period themselves until the actual analysis: cooled, darkened and poisoned with mercury chloride. It has been accepted since the early 1980s that the addition of a toxin to sediment trap samples prevents bacterial or microbial degradation of the material (see e.g. Honjo et al., 1982). Metfies et al. (2017) have even found that PCR-based molecular genetic analyses are possible in sediment trap samples from long-term moorings if the samples have been poisoned with mercuric chloride. Therefore, we are very confident that the samples are not significantly altered when using mercuric chloride. (see corrections lines 182-191)*

Explain how the filtered material is recovered from the polycarbonate filters after drying.

- *The dried sediment trap material forms a so-called filter cake (thick particle layer) on the polycarbonate filters. This material can be easily removed with a spatula from the PC filter after the drying process. This also ensures that no PC filter material mixes with the sample; the sample material is not affected by the PC filter. (see lines 221-230)*

Vertical trends of <1mm POC fluxes are investigated by applying Martin curve fits to average profiles. To what extent is information lost due to this averaging of profiles? What sense does it have to fix the MLD at 10 m? In the Martin approach the MLD is set at 100m depth. Is there any reason to choose a shallower MLD depth? No information is given on observed MLD and its variability. Martin curves do not provide useful information when extrapolated to shallow depths (<100m). In that sense the discussion at page 15 (lines 320-330) needs to be reconsidered.

- *The Martin curve describes the decreasing POC flux with depth below the MLD. It is one of the most often used curves of its kind and the MLD is adjusted to the current depth. Giering et al. (2014) for example, use an MLD of 50 m. Coastal upwelling areas are in turn characterized by a shallow MLD as warmer surface water is transported offshore and replaced by cold deep water. Temperature profiles at our long-term mooring in the SBUS off Hondeklip Bay are published in Rixen et al. (2021). These profiles show that the MLD even has a diurnal cycle and is deepest with around 18 m at 8:00 in the morning (see Fig. 5a). An MLD of 10 m is therefore assumed to be*

*representative for the region (see new Fig. 9 for clarification of MLD in the nBUS and sBUS, and discussion thereafter).*

Discussion about delivery of POC to the sediments (pp14-15) is focussed on the passive <1mm POC flux. But what about the larger stuff ? Unclear how it is taken into account.

- *All particles, including passive particles larger than 1 mm, fall into the drifter cups. The size fraction of passive particles is not further classified in this paper. By sieving the sample to divide it into particles larger and smaller than 1 mm, the large amorphous aggregates disintegrate. Organisms are classified as active and passive as mentioned above. (see modified method section and lines 192-198)*

Specific comments:

Figures and legends are incomplete. Fig 1 does not show Hondeklip, Cape Columbine, nor mentions units for POC concentration. Specify these these are sediment POC contents; nBUS and sBUS delimitation should be indicated. Fig 2 legend should provide a reference for the O2 data of cruise SO285. Fig 3 legend lacks information about the considered surface areas, and time period used for averaging, red and black dots are not identified; the graphs should have markers indicating the timing of the 3 cruises; there is no reference for the used data set. Fig 6 specify that % of zooplankton is biomass % . Fig 9 is not necessary in my opinion and if shown should indicate error bars on the measures POC fluxes.

- *Many thanks for the valuable advice. We will revise the illustrations mentioned in line with the suggestions and improve and adapt them in a revised version of this manuscript. (Fig. 1 was extended with labels and units for the displayed POC, Fig. 2 was extended with sources for the O2 concentrations and a legend so was Fig. 3, captions where modified see lines 145-148, Fig. 6 was modified as mentioned, Fig. 9 was replaced)*

Line 183: the value of 1.8 to convert POC to OM comes out of the blue

- *The conversion from POC to OM comes from the literature (e.g. Anderson, 1995; Francois et al., 2002) and has been used for decades. We have added the reference for clarification. (see lines 233-234)*

Line 210: analysis of the main components of the dried material 'as described before'.. I don't think this was described previously in the text.

- *Indeed, we have missed out on briefly discussing the measurement methods (flash combustion). We will make up for this in the revision (see lines 221-227).*

Lines 245 tyo 252: Not clear why passive, active POC fluxes are averaged over the water column when comparing sites. Onlmy comparing fluxes at given horizons between sites would make sense.

- *That's good, and that's what we did. The drifter traps were mostly at the same depth. According to the Martin and all other curves, the POC fluxes decreases predictably with depth. Comparing the results of the individual drifters shows that this is rarely the case. This raises the question of whether the curves and our idea of particle flux are not all wrong? However, if one takes mean fluxes at corresponding depth levels, a Martin curve*

*can be fitted to the mean values. This shows that the particle has a pronounced variability, but on average follows the Martin curve. In our opinion, the comparison of individual measurements in such a variable system is even misleading, they can be better described if they are recorded on average. And this is what we have done for the NBUS and SBUS. By comparing the active POC flux over the whole nBUS and sBUS over several years and seasons brings also the potential of picturing general differences in biomass (see general details lines 268-273)*

Line 286: Zooplankton abundance is mentioned here. Nothing has been set before in the methiods section about his.

- *In this paragraph, we discuss the percentage of biomass in the drifters. The abundance of zooplankton mentioned here refers only to that determined in Verheye et al. (2016) and Bode et al. (2014). As copepods make up the largest group of zooplankton in the drifters in terms of percentage and numbers. These data originate from the quantitative counting and weighing of the fraction >1mm from the drifters. (see lines 309-314)*

Line 297: Possible occurrence of active swimmers. This important issue should have been tackled before in the method section.

- *This comment is important as it has always been difficult to quantify the proportion of active swimmers in sediment trap samples. Work such as Weldrick et al. (2021) provides approaches on how to deal with active swimmers in drifting sediment trap samples, this procedure is similar and comparable to the approach in this paper. In the methods section, a paragraph will be added to do full justice to this controversial topic and to specify how active swimmers were handled in this work. (modified method section see lines 203-213)*

Line 332: Assuming a mean water depth of 150m ... ? Unclear what the purpose is. Holds for the moored traps only?

- *The question we ask ourselves here is how much POC reaches the sediment surface. Based on the derived Martin curve, this can be calculated if the water depth is known. In order to obtain an averaged value for the continental shelf, we have assumed a water depth of 150 meters. (see line 359)*

**Referee #2**

*We would like to thank the anonymous reviewer for his valuable comments and suggestions for corrections. In the following, we would like to respond to the points addressed in detail.*

Comments:

Figure 1. Please clarify what the contour lines represent; The green diamonds are difficult to distinguish in the figure.

- *The contour lines represent the partitioning in 2.5 % steps of the organic carbon content. The scale is spread from 0-15 percent POC content with a contour line every 2.5 percent for better visualization. We will refit this figure according to the suggestions. (see modified Fig 1 and caption lines 92-94)*

Figure 2. The contour labels are too small; I suggest enlarging them and adding benthic topography to the figure. Since Figure 2 appears to be from one of the research cruises, it should be placed in the results section. Introducing cruise results without first providing details on the cruise, sampling, and methodology is confusing.

- *The contour lines and labelling will be adapted for better visualization. As the introduction section deals extensively with the OMZs in the nBUS and sBUS and the existence of these is known from previous work, it is important that this figure appears as early as possible. The data used here will be labelled and a note added where exactly the data is described in more detail in order to minimize any potential confusion. (Fig 2 remains in introduction section since the $O_2$ concentration in the BUS is crucial for the depicted assumptions later about the POC concentrations, benthic topography was added so were bigger contour labels and a detailed data source)*

Figure 3. Describe the data source in the figure caption. Are these results from the cruise? Clarify what the black and red lines represent in the figure caption.

- *The legend will be added for better understanding. The red dots reflect the sBUS and the black dots reflect the nBUS in both graphics. The data used here originates from satellite data and the respective references are given the method section (line 218 – 222). However, the figure will be adapted and extended to include the source of the data. (see modified Fig.3 and caption lines 145-148)*

Lines 145–148: It is not clear which data are from satellites. Is it Figure 3? Please provide clarification. Additionally, clarify whether the data aligned with satellite data are from the author's observations or previous observation results. This requires either a reference citation or a figure citation. General comments again: The author should differentiate more clearly between the introduction and results. Results from this study should be presented in the results section after introducing the methodology. Please consider adjusting the paper structure.

- *This sentence describes the satellite data shown in Figure 3 and obversions along the Namibian monitoring line off Walvis Bay line. The reference to the observations (Louw et al. 2016) is given after the following sentence in lines 149-150. It will be brought forward to the sentence written in lines 151-154. For better readability and to improve the flow of the text, Figure 3 will be better integrated into the text and described in more detail (see paragraph lines 149-159 with additional sources and integration of Fig.3)*

Line 148 – 153: Again, Results from this study? Previous studies? Citations?

*These lines comprise two sentences:*

- *1) The highest concentrations of chlorophyll were found in these transitional phases, with clear maxima at the beginning and end of the summer between November and January, and March and April, respectively.*

    *2) Averaged over the two subsystems, the primary production derived from satellite data follows the seasonal pattern of chlorophyll concentration off Walvis Bay in both subsystems in so far as that primary production is lower on average in winter than in summer.*

- *The first sentence describes data/results obtained from Louw et al. 2016 and the second sentence refers to primary production rates shown in Figure 3. Since none of these data are ours but describe the working area, we are convinced that they belong into the section "working area". To clarify this issue, we have changed the two sentences as follows:*

    *"Highest concentrations of chlorophyll were found in these transitional phases, with clear maxima at the beginning and end of the summer between November and January, and March and April, respectively (Louw et al. 2016) "(see lines 152-154).*

    *Averaged over the two subsystems, the primary production derived from satellite data (see Fig. 3) follows the seasonal pattern of chlorophyll concentration off Walvis Bay in both subsystems in that primary production is lower on average in winter than in summer (average over 1981-2023). (see lines 155- 158)*

Line 232- 235: Statistical support? Also, please clarify the sentence. Are you indicating that the primary productivity (PP) shows no differences between nBUS and sBUS, or that the PP measurements for both nBUS and sBUS align well with the satellite data?

*These lines include the two sentences:*

- *1) They thus fell below the average primary production rates, which were 2505.3 mg m-2 day-1 in the nBUS and 2089.6 mg m-2 day-1 in the sBUS (Fig. 3b).*
- *2) Overall, the primary production derived from the satellite data largely fell within the range of primary production rates determined during research cruises (140 – 8830 mg m-2 day-1) and hardly revealed any difference between the nBUS and sBUS (Barlow et al., 2009).*
- *The statement of the second sentence is that the mean satellite-derived primary production rates of 2505.3 mg m 2 day-1 in the nBUS and 2089.6 mg m 2 day-1 in the sBUS are within the range of those measured during cruises (140 – 8830 mg m 2 day 1). Is not clear what kind of statical support is required to prove that numbers of 2505.3 and 2089.6 are > 140 and < 8830. We, however, agree that the second part of the second sentence is not sufficiently and clearly worded. It states that available field observations reveal no difference between the nBUS and sBUS. Due to lack of data, this part can be deleted as it adds no relevant information to what is already shown in Fig. 3b. (see lines 248-252).*

Figure 6: I suggest using bar plots with error bars instead of scatter plots with error bars to better address the comparison. Label the bar plots with the mean and standard deviation. Additionally, the percentage of the zooplankton group in the trap appears to be negative—please explain this observation. Furthermore, please provide more detailed information in the caption. Specify what the error bars mean.

- *The suggestion to convert the scatter diagrams into bar charts in order to obtain a better overview is very helpful and will be implemented. As the percentage of weight is very low for some cases, it looks in the scatter plot as if the error extends into the negative area of the chart. This is not the case, all observations that appear in this diagram are positive. The improved visualization with the bar charts can hopefully counteract this optical illusion. The error of the scatter points shown here refers to the average standard deviation of all samples summarized in the zooplankton clades. This information is added to the graph to provide more clarity. (see modified Fig. 6 and modified caption lines 265-267)*

Lines 247-250: Please use statistical analysis instead of visual approximation.

- *Statistical factors are added to the figures mentioned in the text to better highlight their significance. (see lines 268 and 273)*

Lines 251-252: Is it the average of the entire column?

- *The values given in this section refer to the average POC values over the entire water column distributed in the NBUS and SBUS of all drifters that were deployed. This will be highlighted again as a further addition. (see line 269)*

Line 254: This sentence is unclear. Why are you comparing the depth of 64 meters in the nBUS with the depth of 100 meters in the sBUS? Please provide further clarification and explanation for this comparison.

- *We compared data obtained from long-term sediment trap experiments in the two subsystems. These experiments have been conducted at the given water-depth so that the deployment depths of 64 and 100 m are fix (see lines 277-279). Considering the general trends and uncertainties regarding the current understanding of the decline of POC fluxes with water-depth, one could argue that effects caused by depth-differences of 36 m are negligible. However, in a second step we considered depth difference as shown in Figure 8.*

Line 293- 297: These statements need citations.

- *The source cited in the text, Landry and Steinberg 2017, which describes the relationship between zooplankton and the marine C cycle in detail, is supplemented by publications by Boyd et al., (2019), Cavan et al., (2017), Ducklow et al., (2001) and Miles (2018) and Moigne (2019) which show the direct and indirect influence of zooplankton on particle transport and its effect on the biological carbon pump through processes such as grazing etc. (see extended citation and sources in lines 315-322)*

Line 306- 307 Any arguments to support his adjustments? Or it is just a random number?

- *Yes, it fits best to the long-term data and the differences between these two factors in comparison to the measures data is shown in Figure 8b. (see also lines 333-338)*

The discussion in this paragraph about the Mixed Layer Depth (MLD) is unclear. The author should better organize the reasoning. Why use a fixed minimum MLD not varied MLD from cruise or model data? Why does setting MLD to 1 not match primary production (PP) derived from satellite data (if my understanding is correct)? Is it still reliable to calculate the ratio using export production to primary production when they do not match? The author should clearly highlight these points.

- *We agree that this part is a bit confusing, W deleted the discussion about potential consequences of a MLD of < 1 m. We simply used a MLD of 10 m because this agrees to observations of the MLD at the mooring sites in the nBUS (see Rixen et al 2021- Figure 5) and the nBUS. To prove the latter the additional Figure 9 was integrated. Since the minimum MLDs have been around 1 m in summer and varied between 15 and 30 m in winter, we chose an MLD of 10 m to calculate the mean POC fluxes by using the Martin curve. We could perhaps have chosen 15 m, but that would have made almost no difference to the results.*

  *The paragraph has been changed as follows:*

- *When fitting the determined POC fluxes to the curve, we assumed a mixed layer depth (MLD) of 10 m. This corresponds to the shallowest deployment depth of our traps (see Table 3), and it roughly corresponds to observations of the MLD near our mooring sites in the sBUS (Rixen et al., 2021) and nBUS. Compared to the primary production rates of 2089 mg C m 2 day 1 (sBUS) and 2505 mg C m 2 day 1 (nBUS), the 10 m export production of 1117.2 mg C m 2 day 1 suggest an f-ratio (=export production/primary production) of approximately 0.4 to 0.5, which is characteristic of highly productive systems (Eppley and Peterson, 1979). The mean passive POC flux in the BUS thus appears to follow the general attenuation equations, which means that we have performed a sufficiently high number of experiments that allowed us to recognize this general rule despite the large spatial and temporal variability. (see lines 344-349)*

Line 309: This needs clarification. What is the measured POC? Is it only the passive POC, or does it include both passive and active POC? Or are these data from other research? It does not make sense to compare the measured overall POC (active + passive) with the calculated POC flux derived solely from passive flux using the Martin equation.

- *This paragraph refers to the passive POC content only which was analyzed and calculated from samples of our drifters. The active POC content is not considered in this paragraph. Therefore, the data can be displayed and calculated using the Martin curve and the Pearson correlation is significant (see lines 347-349).*

Figure 9:  Need more captions.

- *We have decided to replace Figure 9 with a figure showing the MLD depth in the nBUS and sBUS based on measured SST data from CTD profiles due to the modifications in the paragraph on MLD. No corrections possible.*

Line 350: "They show" – Who are "they"? The language in this discussion section is too casual. For example, phrases like "Bearing in mind" and "as mentioned before" should be replaced with

more formal language. The author should organize their arguments logically and present them in a formal manner. I strongly suggest rewriting this paragraph.

- *In order to raise the scientific level of this paragraph accordingly, the colloquial passages will be modified and the logical structure of the paragraph will be revised so that the reading flow is not disturbed. (see lines 375-396)*

Lines 348-349: The interaction between active POC and passive POC is not well explained. The explanation is unclear and difficult to follow. I suggest adding a comparison figure to illustrate the mechanisms involved.

- *To illustrate the complex relationships between the active and passive POC flux, a system diagram of the main underlying processes will be added. We have attached a system graphic to illustrate the important processes. (see new Fig 10 which shows the underlying described processes in the nBUS and sBUS)*

Lines 349-350: Since the passive POC flux is consistent in the deep layer of both regions, does this imply that the increased production of fecal pellets in the sBUS should be converted to dissolved inorganic carbon (DIC) in the shallow water? Did you measure water column DIC, pH, or any other inorganic carbonate parameters?

- *During the research cruises in which the drifters were used, parameters of the inorganic C-cycle were also measured in detail (see Siddiqui et al., 2023). A comparison of the drifter results with the parameters of the inorganic C-cycle is beyond the scope of this paper, as it is not straightforward to convert the fraction of fecal pellets to DIC, as this quantification must be preceded by a complex processing approach to determine the exact fraction of fecal pellets (Leigh et al., 2024). In a highly dynamic system such as the BUS with strong DIC and TA gradients mixing processes largely mask the comparably small effect of organic matter respiration on the DIC concentrations. Nevertheless, the coupling of abiological and biological C-pump is an interesting factor that should definitely be investigated in more detail in further work. (see lines 388-396)*

Lines 353-355: This discussion needs more support. The mud belt event is not discussed in sufficient detail, and the correlations between this event and your findings are unclear. Please expand and provide a more thorough discussion of these last couple of sentences

- *Mud belt is not an event but a region off Namibia characterized by high organic carbon concentrations as shown in Fig. 1 and explained in lines 86 – 90 (see also Emeis et al. 2018) (see additional sources line 396, with Fig. 10 for the mud belt location and more details)*

**New References:**

Anderson, L. A. (1995) On the hydrogen and oxygen content of marine phytoplankton. Deep-Sea Research 1, 42(9), 1675–1680.

Boyd, P. W., Claustre, H., Levy, M., Siegel, D. A., & Weber, T. (2019). Multi-faceted particle pumps drive carbon sequestration in the ocean. Nature, 568(7752), 327–335. https://doi.org/10.1038/s41586-019-1098-2

Cavan, E. L., Henson, S. A., Belcher, A., & Sanders, R. (2017). Role of zooplankton in determining the efficiency of the biological carbon pump. Biogeosciences, 14(1), 177–186. https://doi.org/10.5194/bg-14-177-2017

Ducklow, H. W., Steinberg, D. K., & Buesseler, K. O. (2001). Upper ocean carbon export and the biological pump. Oceanography, 14(SPL.ISS. 4), 50–58. https://doi.org/10.5670/oceanog.2001.06

Francois, R., Honjo, S., Krishfield, R., and Manganini, S. (2002) Factors controlling the flux of organic carbon to the bathypelagic zone of the ocean, Global Biogeochemical Cycles, 16.

Honjo, S., Manganini, S. J., and Cole, J. J. (1982) Sedimentation of biogenic matter in the deep ocean, Deep-Sea Research, 29, 609-625.

Le Moigne, F. A. C. (2019). Pathways of Organic Carbon Downward Transport by the Oceanic Biological Carbon Pump. Frontiers in Marine Scie, 6(October), 1–8. https://doi.org/10.3389/fmars.2019.00634

Lee, C., Hedges, J., & Wakeham, S. (1991). Technical problems with the use of sediment traps - preservation, swimmers, and leaching. In P. Wassmann, A.-S. Heiskanen, & O. Lindahl (Eds.), Symposium Proceeedings - Sediment trap studies in the nordic countries (Vol. 2, pp. 36–48). Kristineberg Marine Biological Station. Nurmijavi, ISBN-952-90-2844-X.

Metfies K., Bauerfeind E., Wolf C., Sprong P., Frickenhaus S., Kaleschke L., Nicolaus A., Nöthig E.-M. (2017) Protist Communities in Moored Long-Term Sediment Traps (Fram Strait, Arctic) – Preservation with Mercury Chloride Allows for PCR-Based Molecular Genetic Analyses. Front. Mar. Sci. 4:301. doi: 10.3389/fmars.2017.00301.

Miles, M. (2018). The Biological Carbon Pump: Climate Change Warrior. Berkeley Scientific Journal, 23(1). https://escholarship.org/uc/item/7cg4n7p8.

Siddiqui, C., Rixen, T., Lahajnar, N., Van der Plas, A. K., Louw, D. C., Lamont, T., & Pillay, K. (2023). Regional and global impact of CO2 uptake in the Benguela Upwelling System through preformed nutrients. Nature Communications, 14(1). https://doi.org/10.1038/s41467-023-38

Smetacek, V., von Bröckel, K., Zeitzschel, B., Zenk, W. (1978) Sedimentation of particulate matter during a phytoplankton spring bloom in relation to the hydrographical regime. Marine Biology 47, 211-226.

Weldrick, C. K., Makabe, R., Mizobata, K., Moteki, M., Odate, T., Takao, S., Trebilco, R., & Swadling, K. M. (2021). The use of swimmers from sediment traps to measure summer community structure of Southern Ocean pteropods. Polar Biology, 44(3), 457–472. https://doi.org/10.1007/s00300-021-02809-4

---

## Referee Report (RR1)

Abstract:

Specify what you mean by 'ambivalent' nature of the OMZ

Introduction:

Line 171: here depths are trap depths, not water depths.

Line 123: '..in the shadow of oceanic fronts ..' can you specify?

Methods section:

Take Table 3 closer to line 176.

The transfer of dried material from polycarbonate filters to agate mortar, may not be 100% efficient for the small particles fraction. Have you an estimate of this transfer efficiency?

Results section:

Reposition the Figure 4 showing with POC fluxes. Now it appears in a section were PP data are shown. Move the figure closer to line 281? Specify in legend of Fig.4 that data shown are for moored traps.

Discussion section:

Line 281: 'However, our time series _in the sBUS_ is still too short ..'

Line 325: ..'the commonly used equation for describing the POC flux ..' refer to Eq. 1 and Martin et al.

Line 329: the choice of taking 10m as the MLD seems somehow arbitrary considering that Temp profiles indicate MLD's between 30 and 14m. For instance using an MLD of 14m instead of 10m will increase Fz values by some 30%. Can you convince the reader of you choosing a 10m MLD? FMLD values should be shown without decimals (Lines 331, 344), considering the variabilities involved.

Line 349: '.. despite the enormous deviation..' replace by 'despite the large variations'

Figure 9 legend: mention M153 is during summer; 4th line: .. that the MLD _at St7_ showed ..

Figure 10 legend: 2nd line: '.. which is identical' replace by 'which is similar' (idem for Line 381)

Lines 381 – 383: sentence is unclear. Reformulate please.

Summary:

Line 406-407: '..and a much higher zooplankton abundance _in the sBUS_ implies that ..'

---

## Author Response (AR3)

**Biogeosciences: egusphere-2024-700**

The influence of zooplankton and oxygen on the particulate organic carbon flux in the Benguela Upwelling System

Luisa Chiara Meiritz, Tim Rixen, Anja Karin van der Plas, Tarron Lamont, and Niko Lahajnar

**Reviews and Comments**

01. October 2024

We are once again very thankful to the editor and to the reviewers for their valuable comments and suggestions for the improvement of our study. We have followed up all comments in detail and have revised the manuscript according to the suggestions and comments. All changes are marked in blue in the current version to make it easier to follow the differences to the original. We have addressed the individual comments of the reviewer as described below.

**Abstract:**

Specify what you mean by 'ambivalent' nature of the OMZ

> *We have rewritten the last sentence of the abstract to make the statement clearer (lines 32-34)*

**Introduction:**

Line 171: here depths are trap depths, not water depths.
> *(changed; line 171).*

Line 123: '..in the shadow of oceanic fronts ..' can you specify?

> (*specified in lines 121-122*)

**Methods section:**

Take Table 3 closer to line 176.

> *Table 3 moved as close as possible in this early stage of the manuscript. Positioning might change during editorial handling of the journal*).

The transfer of dried material from polycarbonate filters to agate mortar, may not be 100% efficient for the small particles fraction. Have you an estimate of this transfer efficiency? Explain how the filtered material is recovered from the polycarbonate filters after drying.

> *We have now explicitly stated how the filters were treated. All filters were thoroughly visually inspected to ensure that no sample material larger than 0.45 µm (mesh size of the polycarbonate filter) remained on the filter surface. This ensures that virtually blank filters were left behind and no particles on the filters remained untreated. (see lines 225-227)*

**Results section:**

Reposition the Figure 4 showing with POC fluxes. Now it appears in a section were PP data are shown. Move the figure closer to line 281? Specify in legend of Fig. 4 that data shown are for moored traps.

> *We have swapped the figure numbering for Figures 4, 5 and 6 so that they match the text passages better. Former Figure 4 (now Figure 6) has been moved close to line 281. The figure caption of Figure 6 explicitly states now that data are derived from mooring deployments.*

**Discussion section:**

Line 281: 'However, our time series in the sBUS is still too short ...' (*changed; line 282*)

Line 325: […] we have selected the commonly used equation (Eq. 1, Martin et al. 1987) for describing the POC flux… (*changed; lines 326-327*)

Line 329: The choice of taking 10m as the MLD seems somehow arbitrary considering that Temp profiles indicate MLD's between 30 and 14m. For instance using an MLD of 14m instead of 10m will increase Fz values by some 30%. Can you convince the reader of you choosing a 10m MLD? FMLD values should be shown without decimals (Lines 331, 344), considering the variabilities involved.

> *We have now explained why we decided on an MLD of 10 m and that this value matches the observations during the different seasons. 'The selected MLD as well as the determined FMLD and 'b' values were then used to calculate the POC fluxes using the Martin curve and the water depth. The calculated and measured POC fluxes correlated with each other (r = 0.925, n = 31) and showed the best agreement at an assumed MLD of approximately 10 m. The values for FMLD and 'b' obtained from the curve fitting were 1117 mg C m$^{-2}$ day$^{-1}$ and -0.74 (Fig. 8). Temperature profiles obtained during our expeditions at our long-term sediment trap sites showed that MLDs in the nBUS and sBUS varied between approximately 30 m and 15 m during the austral spring (Fig. 9a). In late summer, the mean depth varied between about 1 m and 14 m (Fig. 9b), showing that an MLD of 10 m, as assumed for the curve fitting, is within the range observed during our expeditions.' (lines 329-336)*

Line 349: '.. despite the enormous deviation..' replace by 'despite the large variations' (*changed; line 334*)

Figure 9 legend: mention M153 is during summer; 4th line: .. that the MLD *at St7* showed .. (*changed; lines 358 and 361*)

Figure 10 legend: 2nd line: '.. which is identical' replace by 'which is similar' (idem for Line 381) – (*changed; line 374*)

Lines 381 – 383: sentence is unclear. Reformulate please.

> *We have reworded this passage. 'As mentioned above, the attenuation coefficient 'b' is an expression of the strength of remineralisation. According to our results, 'b' is similar in both subsystems, which leads to the assumption that the effect of a lower oxygen concentration on 'b' in the nBUS is comparable to the effect of zooplankton on 'b' in the sBUS.' (lines 386-388)*

**Summary:**

Line 406-407: '..and a much higher zooplankton abundance _in the sBUS_ implies that ..
(*changed; line 411*)